# Advances and Challenges in Drone Detection and Classification Techniques: A State-of-the-Art Review

**DOI:** 10.3390/s24010125

**Published:** 2023-12-26

**Authors:** Ulzhalgas Seidaliyeva, Lyazzat Ilipbayeva, Kyrmyzy Taissariyeva, Nurzhigit Smailov, Eric T. Matson

**Affiliations:** 1Department of Electronics, Telecommunications and Space Technologies, Satbayev University, Almaty 050013, Kazakhstan; k.taisariyeva@satbayev.university (K.T.);; 2Department of Radio Engineering, Electronics and Telecommunications, International IT University, Almaty 050040, Kazakhstan; 3Department of Computer and Information Technology, Purdue University, West Lafayette, IN 47907-2021, USA; ematson@purdue.edu

**Keywords:** unmanned aerial vehicles (UAVs), UAV detection, drone detection, drone identification, UAV identification, UAV classification, drone classification, drone localization, detection technologies, radio frequency (RF), radar, acoustic, visual, sensor fusion, drone incidents, drone threats, machine learning based drone detection, deep learning based UAV identification

## Abstract

The fast development of unmanned aerial vehicles (UAVs), commonly known as drones, has brought a unique set of opportunities and challenges to both the civilian and military sectors. While drones have proven useful in sectors such as delivery, agriculture, and surveillance, their potential for abuse in illegal airspace invasions, privacy breaches, and security risks has increased the demand for improved detection and classification systems. This state-of-the-art review presents a detailed overview of current improvements in drone detection and classification techniques: highlighting novel strategies used to address the rising concerns about UAV activities. We investigate the threats and challenges faced due to drones’ dynamic behavior, size and speed diversity, battery life, etc. Furthermore, we categorize the key detection modalities, including radar, radio frequency (RF), acoustic, and vision-based approaches, and examine their distinct advantages and limitations. The research also discusses the importance of sensor fusion methods and other detection approaches, including wireless fidelity (Wi-Fi), cellular, and Internet of Things (IoT) networks, for improving the accuracy and efficiency of UAV detection and identification.

## 1. Introduction

In recent years, as a result of the ongoing development of technology, micro unmanned aerial vehicles (UAVs), more often referred to as drones, have seen improvements in their technical capabilities and an expansion in their range of applications. Due to their ability to fly long-range distances and their compact size, mobility, and payload options, the potential applications for drones have expanded from personal to military usage. Drones play a significant role in modern life as a result of their low cost and ease of usage in many sectors of daily life, from official government work like border security and wildfire surveillance to civilian private sector work such as first aid, disaster management, monitoring of crowded places, farming, delivery services, internet communications, and general filmmaking [1]. Therefore, drone technology’s democratization has resulted in broad acceptance, making the sky more accessible than ever before. Along with the benefits, the increased use of drones has created substantial issues in ensuring the security, privacy, and safety of both airborne and ground-based organizations. Effective drone detection and classification systems have become a key concern for regulatory agencies, security services, and aviation authorities all over the world. The central goal of this review paper is to investigate the advancements in drone detection and classification algorithms that have evolved as a result of these issues. It aims to present a comprehensive overview of the state-of-the-art technologies that are now in use or are being developed. This involves an investigation of several detection modalities, such as radar, radio frequency (RF) analysis, acoustic sensors, and camera sensors, and the fusion of these systems into comprehensive detection networks. Moreover, the paper covers the importance of sensor fusion methods and other detection approaches, including Wi-Fi fingerprinting, 5G, and IoT networks, for improving the robustness of UAV detection and identification systems.

The key objectives of this review paper are to:-describe the most recent drone incidents and threat categories;-identify and describe the main approaches utilized for drone detection and classification;-analyze the advantages and limitations of various techniques;-emphasize the interference factors in real-world scenarios for individual sensor modalities and suggest additional integrated approaches.

In order to enforce a consistent set of rules, drone regulations are now being prioritized throughout all nations. But as technology has advanced, it is possible to create non-registered, bespoke drones that might fly in prohibited areas [2]. Numerous incidents and events involving the usage of drones flying near vital infrastructure, over restricted locations, or during important public meetings have been recorded in the media in recent years [1]. To demonstrate the necessity for a drone detection system, below we briefly discuss a couple of the threat categories in the context of concrete incidents.

### 1.1. Significance of UAV Detection: Drone Threat Categories and Incidents

Recently, the horizon of drone use has expanded rapidly from military to smart agriculture. As the use of drones spreads across industries, the ability to detect and identify them effectively becomes increasingly critical in order to avoid illegal or harmful drone activity. The motivation for this research derives from the increasing use of drones and the hazards they bring to privacy, security, and safety. This research is significant because it provides a contemporary and relevant overview of drone detection technologies, which are vital for national security, privacy, and safety. Drones serve as delivery vehicles for various payloads, such as GPS devices, infrared cameras, batteries, and sensors. These drones often use high-energy batteries, such as lithium batteries, to provide a flight duration of 20–40 min. Drones that can fly several kilometers away from their operator and stay in the air for thirty minutes may be purchased for around USD 1000. This therefore gives terrorist organizations the ability to utilize drones for both the delivery of explosives or toxic materials and for the surveillance of and gathering of data from targets [3]. Therefore, drones with long-range flight capability, heavy-payload-carrying capacity, and wind and weather resistance may serve as a means for transporting dangerous goods.

In order to determine proper anti-drone system goals, we list the factors of unmanned aerial vehicles that represent the greatest threats to society (Figure 1).

*Drone attacks*: Due to the fact that unmanned aerial vehicles are capable of carrying explosives as well as biological and chemical weapons, drone attacks fall under the first category of possible threats. These explosives might be used to attack a variety of targets, such as specific individuals, public institutions, business organizations, and even whole nations [4].

*Illegal smuggling*: Smuggling is the second-leading threat category for drone use. For border patrols and jail staff, drone drug smuggling has grown to be a serious issue. Sometimes weapons or other illegal items are smuggled beyond the reach of ground-based security. Border locations have a wide range of weather; therefore, smuggling drones need to be able to operate in adverse weather conditions [1].

*Drone espionage threats*: Drones with strong cameras may also be used to spy on people, businesses, and governmental institutions from a distance. Despite privacy claims, this worry might be a drone hazard, such as a privacy invasion or espionage [3].

*Drone collisions*: Accidentally or purposefully launching a remote-controlled drone near an aircraft or in its flight path might threaten the safety of the crew and passengers and might damage property [3].

In 2023, a surge of drone incidents has been continuing to hit the news, highlighting the need for strong counter-drone solutions and rational rules for using small UAVs. These incidents ranged from smuggling and illegal drone intrusions to collisions and technological failures and affected a variety of industries. The majority of rogue drone actions occurred in sensitive areas such as airports, border crossings, prisons, communities, and neighborhoods. The total number of reported drone incidents since January 2023 is around 150 [5]. Incidents related to the delivery of drug contraband at borders and surveillance of border patrols using drones make up 38% of total drone incidents (Figure 2). Especially, drone smuggling has increased across the India–Pakistan border and is occurring several times a month [6] (Figure 3). Approximately 84% of reported drone smuggling incidents took place across the India–Pakistan border, 12% across the Jordan–Syria border, and 4% of reported drone smuggling incidents were attempted at the Israel–Lebanon border. As well, drone smuggling and the delivery of illegal items into prisons by visitors or corrupt jail personnel have always been issues. A total of 28% of all reported drone incidents involved drones delivering illegal items and drugs into prisons. More than half of the drone smuggling cases occurred in prisons in the United States and Canada; the rest were recorded in prisons in the United Kingdom. Drone smuggling incidents have also been reported in prisons in Spain, France, India, and Ecuador. In 2023, apart from the prison and border sectors, airports also faced some challenges due to illegally launched drones. Therefore, 18% of reported drone incidents were related to drones flying close to runways. According to an analysis of global drone incidents for 2023, Dublin Airport was the most affected by drone incidents. Also, thousands of passengers at several airports in the UK and the USA were inconvenienced due to the fact that drones flew too close to planes and caused temporary suspensions of flights and flight delays. As a result of the illegal use of weaponized drones, different communities and neighborhoods, mostly in the USA, India, Israel, and the UK, have faced various challenges of violence [6]. Figure 3 illustrates the number of different drone threats for each month, starting from January 2023 until the beginning of December 2023.

A list of the significant drone incidents is provided in Table 1.

The countries with the highest number of drone occurrences based on analysis of recent drone incidents reported by the press worldwide are: India, the USA, the UK, and Canada. Most of the drone incidents were related to illegal items being delivered into prisons using drones and drone flight over airports or near borders or restricted zones. The majority of drone models involved in recent drone incidents were the DJI Mavic 2, DJI Phantom 4, DJI Mavic 3, etc. All of the above-mentioned incidents highlight the necessity for UAV detection systems to stop unauthorized drone applications. The goal of such a detection system is to identify drones’ existence as well as their location and other details, including their type, direction, and velocity. In light of this, this research problem has recently attracted a lot of interest from both businesses and academia. The research problem is compounded by the rapid evolution of drone technology, which includes their tiny size, low flying altitude, and low radar cross-section. Also, drones may readily enter no-fly zones, and they are more difficult to detect than conventional big air targets. Consequently, detection of suspicious drones becomes more challenging [17].

### 1.2. Key Challenges to UAV Detection

The UAV detection and classification task presents numerous significant challenges, which researchers [18,19] and business experts are currently addressing. Some of these challenges include:

*1. Size and speed diversity*: UAVs are available in a range of sizes, from tiny consumer drones to bigger commercial or military-grade aircraft. The majority of tiny UAVs fly at speeds of 15 m per second or less. However, large-sized UAVs have a higher top speed of up to 100 m/s [18,19]. Since various UAVs may have distinct flying characteristics and shapes, these size and speed variances make UAV detection and classification tasks more challenging.

*2. Dynamic behavior and similarity to other flying objects*: It is challenging to monitor and precisely identify UAVs because of their high speed and unpredictable flying patterns and behaviors. The difficulty for detection and classification systems increases when dealing with UAVs’ dynamic nature. As well, the large similarity of drones to other objects, such as birds or airplanes, makes it difficult to distinguish them from other flying objects. Therefore, a drone detection system must be fast enough not to miss a drone in video frames and yet accurate enough not to confuse the drone with another flying object. This is such an urgent problem that a drone vs. bird challenge is organized each year [20].

*3. Different detection ranges and altitudes*: UAVs may fly at various altitudes, between a few meters and many kilometers above the ground. A UAV’s range refers to the region from which it may be remotely controlled. While larger drones have a range of hundreds of kilometers, smaller drones have a range of only a few meters. Hence, aerial platforms might be divided into low-altitude and high-altitude platforms. Due to their quick deployment and inexpensive cost, low-altitude platforms are mostly used for malicious purposes. As malicious drones frequently fly at low altitudes, traditional countermeasure systems may be ineffective or even harmful owing to possible collateral damage [18].

*4. Environmental conditions*: Environmental factors that can impair the effectiveness of UAV detection systems include varying weather conditions [17] (such as rain, fog, or wind); urban locations with plenty of obstructions, such as buildings and trees; terrain; background noise; adverse lighting conditions [21]; etc. Severe environmental factors can affect the precision and robustness of sensors, such as radar or optical systems, resulting in false positives or false negatives in UAV identification. Thus, it is an ongoing challenge to adapt and modify detection and classification algorithms to deal with various environmental circumstances.

*5. Limited battery life*: One of the biggest issues that UAVs encounter is their limited battery life. Increasing the battery size of UAVs is not possible since this would increase weight, which is another key consideration. Most research has concluded that lithium–polymer (LiPo) batteries are the most commonly utilized battery type in drones; nevertheless, lithium–iron–phosphate (LiFePO4) batteries are believed to be safer and have a longer life cycle [22,23]. Drones can fly in the air for 30–40 min before landing, and swapping batteries might be troublesome. Temperature, humidity, altitude, and flight speed all have a major influence on drone batteries. However, the charging strategies for these batteries are equally critical. The right charging protocol is crucial for extending battery life and ensuring consistent performance. To mitigate the impact of UAV flight time limitations imposed by limited on-board battery life, it is critical to reduce energy consumption in UAV operations [22].

These challenges drive continuing research into the creation of novel detection and classification methods and the advancement of sensor technology. As well as addressing these challenges, this topic calls for multidisciplinary study and cooperation amongst specialists in artificial intelligence (AI), computer vision, and signal processing. Also, overcoming these challenges can aid in ensuring the security, privacy, and safety of both people and vital infrastructure.

## 2. Drone Detection Technologies

In recent years, many research works have been published to address UAV detection, tracking, and classification problems. The main drone detection technologies are: radar sensors [24,25,26,27,28,29,30,31,32,33,34,35,36], RF sensors [37,38,39,40,41,42,43,44,45,46,47], audio sensors [48,49,50,51,52,53,54,55,56,57,58,59,60], and camera sensors using visual UAV characteristics [61,62,63,64,65,66,67,68,69,70,71]. Based on the above-mentioned sources, the advantages and disadvantages of each drone detection technology are compared in Table 2. In addition, the academic and industrial communities have begun to pay attention to bimodal and multi-modal sensor fusion methods; however, these methodologies are still in the early stages of development and implementation.

### 2.1. Radar-Based Detection

*Principles of radar-based detection:* Radar, which stands for “Radio Detection and Ranging”, is generally considered one of the most trustworthy sensing devices that comes to mind when addressing UAV detection since it has traditionally been utilized for aircraft detection in both military and civilian purposes (such as aviation) [17]. Radar is an electromagnetic technology that employs radio waves to detect and locate nearby objects. Any radar system works on the basis of echo-based measurements and consists of a radar transmitter that sends out short electromagnetic waves in the radio or microwave band, transmitting and receiving antennas, a radar receiver that receives the reflected signals from the target, and a processor that identifies the objects’ attributes. Therefore, radar can calculate important object characteristics, including distance, velocity, azimuth, and elevation [24].

*Types of radar:* Radars come in two varieties: active radars and passive radars. Active radar transmits a signal and then receives the reflected signal to detect objects. Therefore, if the radar sensor illuminates objects, it is defined as active [25]. In contrast, passive radar does not emit any kind of signal; instead, it relies on external signal sources such as natural sources of energy (the Sun and stars) as well as other signal sources, including cellular signals, frequency modulation (FM) radio, etc. Active radars are frequently referred to simply as radars in the literature. If there is a difference between the transmitting and receiving antennas, active radar is said to be bistatic; otherwise, it is referred to as monostatic. An (active) radar transmits either trains of electromagnetic energy pulses or a continuous wave; in the first case, these are known as continuous wave (CW) radars, such as stepped frequency continuous wave (SFCW) radar; in the second case, these are known as pulse radars. Pulse-Doppler radar is an alternative form of radar that combines the characteristics of the two radar systems mentioned above. Depending on operating frequency band designations, radar sensors have several classifications. Radar frequency bands in accordance with Institute of Electrical and Electronics Engineers (IEEE) standards and their typical applications are presented in [25].

Several common varieties of radar are briefly explained below:

*(1) Surveillance radar*: The typical application for this kind of radar is long-range surveillance and detection. It has extensive coverage and can detect UAVs up to a few kilometers away. Radars used for surveillance often operate in the X-band or S-band frequencies and feature an elevated platform to increase the detection range. In [26], the authors proposed a reliable bird and drone target classification method using motion characteristics and a random forest model. Different flight patterns and motion characteristics of birds and drones, such as velocity, direction, and acceleration, extracted from the surveillance radar data were used as the primary features for the classification system.

*(2) Millimeter-wave (mmWave) radar*: The use of mmWave technology in radar systems can be an effective tool for UAV detection due to its abilities to penetrate various weather conditions, improve resolution, and assist in the detection of tiny drones. A mmWave radar uses radio waves with wavelengths ranging from 1 to 10 mm and can detect the presence of drones by measuring the radio waves reflected off their surfaces. In [2], the authors presented a novel drone classification approach using deep learning techniques and radar cross section (RCS) signatures extracted from millimeter-wave (mmWave) radar. The majority of drone classification techniques typically convert RCS signatures into images before doing classification using a convolutional neural network (CNN). Due to the added computational complexity caused by converting every signature into an image, CNN-based drone classification shows low classification accuracy concerning the dynamic characteristics of a drone. Thus, by adding a weight optimization model that can minimize computing cost by preventing the gradient from flowing through hidden states of the long short-term memory (LSTM) model, the authors presented an enhanced LSTM. Experimental results showed that the accuracy of the long short-term memory adaptive-learning-rate-optimizing (LSTM-ALRO)-based drone classification model is much greater than the accuracy of the CNN- and Google-based models.

*(3) Pulse-Doppler radar*: This type of radar emits short radio wave pulses and detects the frequency shift brought on by the motion of an unmanned aerial vehicle (UAV) even in the presence of background noise or interference [27].

*(4) Continuous wave (CW) radar*: This type of radar detects unmanned aerial vehicles (UAVs) by continuously transmitting radio waves and analyzing the frequency shift in the reflected signal [28].

*(5) Frequency-modulated continuous wave (FMCW) radar*: FMCW radar continually transmits an electromagnetic signal with a fluctuating frequency over time and uses the difference in frequencies between emitted and reflected signals to determine the range and velocity of objects [17]. These signals, often known as chirps, vary from CW in that the operational frequency is not changed throughout the transmission [29]. Due to their constant pulsing, inexpensive cost of hardware components, and superior performance, FMCW and CW radars are recommended for use in UAV detection and identification [30].

*Micro-Doppler effect in radar*: Compared to conventional measurements like radar cross section (RCS) and speed, micro-Dopplers created by moving blades can be used as more efficient signatures for detecting radar signals from UAVs. Recent years have seen a significant increase in the importance of the detection and classification of UAVs using the radar micro-Doppler effect [28]. Due to the rotating blades of drones modulating incident radar waves, it is known that drones cause micro-Doppler shifts in radar echoes. Specific parts of an object that move separately from the rest provide a micro-Doppler signature. These signatures may be produced by drones simply by rotating their propeller blades [31]. As well, the drone’s moving components, such as rotors or wings, produce distinctive radar echoes known as micro-Doppler signatures. By examining these signatures, the authors of [32] proposed a novel approach that focuses on developing some patterns and characteristics to identify various small drones based on blade types. The presence or absence of these micro-Doppler shifts, which are produced in the spectra of drones, helps to separate drone signals from background noise like birds or humans. Distinctions between drone types were made using the variations in the micro-Doppler parameters developed by the authors, such as the Doppler frequency difference (DFD) and the Doppler magnitude ratio (DMR). X-band pulse-Doppler radar was used to analyze radar signatures of different small drone types such as multi-rotor drones with only lifting blades, fixed-wing drones with only puller blades, and hybrid vertical take-off and landing (VTOL) drones with both lifting and puller blades. Experimental results demonstrated that for all three types of drones, lifting blades produced greater micro-Doppler signals than puller blades.

Even when using very sensitive radar systems, it is not possible to differentiate between multi-copters and birds with sufficient accuracy using either the physical size or the radar cross section (RCS). Investigations of multi-copters for their micro-Doppler signatures and comparison with those of birds have shown excellent findings in related studies [33,36]. In [33], the authors presented the characteristic radar micro-Doppler properties of three different drone models and four bird species of different sizes obtained by processing a phase-coherent radar system at K-band (24 GHz) and W-band (94 GHz) frequencies. The experimental outcomes clearly showed that there are considerable differences between the micro-Doppler signatures of birds and proved that a K-band or millimeter-wave radar system is capable of detecting drones with excellent fidelity. The findings demonstrated that micro-Doppler signatures such as bird wing beat signatures, helicopter rotor modulation (HERM) lines, micro-Doppler dispersion across the Doppler axis, and rotor blade flash might all be employed as classification features for accurate target recognition. Thus, target classification may be accomplished using the micro-Doppler effect based on the feature “micro-Doppler signatures”. Flying multi-copters can be detected using the radar micro-Doppler signatures produced by the micro-movements of rotating propellers. In the case of multi-copters, rotating rotor blades are the primary source of micro-Doppler signatures, while the wing beats of birds provide these signatures. The authors of [33] demonstrated that drones and birds may be consistently distinguished by analyzing the radar return, including the micro-Doppler characteristic. Further, the work [31] addresses the problem of classifying various drone types such as DJI and Parrot using radar signals at X- and W-band frequencies, Convolutional neural networks (CNNs) were used to analyze the short-time Fourier transform (STFT) spectrograms of the simulated radar signals emitted by the drones. The experimental results demonstrated that a neural network that was trained using data from an X-band radar with a 2 kHz pulse repetition frequency outperformed a CNN trained using the aforementioned W-band radar. In [34], the authors proposed a deep-learning-based technique for the detection and classification of radar micro-Doppler signatures of multi-copters. Radar micro-Doppler signature images of rotating-wing-copters and various other non-rotating objects were collected using continuous wave (CW) radar, and then the micro-Doppler images were labeled and fed to a trained CNN model. Experimental measurements showed 99.4% recognition accuracy with various short-range radar sensors. In [35], the authors examined micro-Doppler data obtained from a custom-built 10 GHz continuous wave (CW) radar system that was specifically designed for use with a range of targets, including UAVs and birds, in various scenarios. Support vector machines (SVMs) were used for performing different classification types, such as drone size classification, drone and bird binary classification, as well as multi-class-specific classification among the five classes. The main shortcoming of the research is the limited conditions for data collection, and the authors hope to address this limitation in their future work.

### 2.2. Radio Frequency (RF)-Based Detection

*Principles of RF-based detection*: As electronic components aboard drones, such as radio transmitters and Global Positioning System (GPS) receivers, emit energy that may be detected by RF sensors, RF-based UAV detection is considered one of the most efficient detection techniques. An RF-based UAV detection system is made up of the UAV itself, a UAV remote controller or ground control station, as well as two receiver units for registering the power of the received signal produced by the UAV. One of the receivers captures the RF signals’ lower band, while the upper band of the RF signals is recorded by the second receiver (see Figure 4). The communication signals between the controller device and the UAV are considered RF signals. Drones often operate on a variety of frequencies. However, the majority of commercial drones communicate with their ground controllers by using RF signals in the 2.4 GHz Industrial, Scientific, and Medical (ISM) frequency band. Usually, the drone’s frequency is unknown, and the RF scanner passively listens to the signals sent between the UAV and its controller [17]. RF scanner technologies that capture wireless signals can be used to detect the presence of UAVs in the target area. Therefore, UAV detection in no-fly zones is most frequently accomplished by intercepting and analyzing the RF-transmitted signals between the UAV and the ground control station. Typically, these signals include up-link control signals from the ground station and down-link data signals (position and video data) from the drone [37].

Recent related studies on the use of RF sensors for UAV detection and classification may be found in [37,38,39,40,41,42,43,44,45,46,47].

*State-of-the-art in RF-based UAV detection, classification and identification:* Classification is one of the crucial parts of supervised machine learning techniques. For a given dataset, the number of labels and classes is the same. The binary classification problem matters when traditional machine learning (ML)- or deep learning (DL)-based UAV detection and identification is performed using only two labels, such as “drone” and “no drone”, “drone” and “bird”, etc. On the other hand, multi-class classification means drone identification based on different models, payloads, number of rotors, and flight modes [37]. Recent related studies on the use of RF sensors for UAV detection and classification may be found in [37,38,39,40,41,42,43,44,45,46,47].

*Drone detection and identification (DDI) systems using ML algorithms*: A novel hierarchical learning approach for efficient detection and identification of RF signals for different UAV types was proposed in [37]. The proposed approach is performed based on several stages, such as problem formulation specifying the system model and dataset [39], a data pre-processing stage including smoothing filters, an ensemble learning approach for solving multi-class classification tasks based on the voting principle between two ML algorithms such as eXtreme Gradient Boosting (XGBoost) and k-nearest neighbors (KNN), and model evaluation based on appropriate metrics. The experiment was performed based on a publicly available DroneRF [39] dataset consisting of 227 segments stored in a comma-separated values (CSV) format. The hierarchical learning approach consists of four classifiers that work in a hierarchical way. The first classifier performs binary classification by specifying the presence of a UAV (in this case, there are two labels: “UAV” and “no UAV”). If a UAV is detected, then the second classifier defines the type of UAV (in this case, there are three labels: Parrot AR, Phantom 3, and Parrot Bebop). If the detected UAV is a Parrot Bebop, then the third classifier is activated and specifies the flight mode of the Bebop drone between four flight modes: on, hovering, flying, and flying with video recording. If the detected UAV’s model is Parrot AR, then the fourth classifier defines the flight mode of AR. The DJI Phantom 3 has only one flight mode. To evaluate the detection system, false negative rate (FNR), false discovery rate (FDR), and F1-score metrics were defined using the values of precision and recall. The experiment outcomes demonstrated that with an increase in the number of classes, the classification accuracy decreased. Despite such a challenge, the authors could reach a classification accuracy of about 99% with their proposed method. The same research was performed in [38], wherein the authors ensured a comprehensive comparison of six ML technologies (XGBoost, adaptive boosting (AdaBoost), decision tree, random forest, KNN, as well as multilayer perceptron (MLP) for RF-based drone identification also using the DroneRF [39] dataset. A two-step experiment, including discrete Fourier transform (DFT)-based feature extraction in the frequency domain and ML-based classification, showed that XGBoost provides state-of-the-art outcomes. Another similar work was conducted in [40], wherein the authors proposed an RF-based machine learning drone detection and identification system that leverages low-band RF signals for communication between the drone and the flight controller. The study was carried out utilizing the DroneRF dataset [39], which is accessible to the public and contains 227 segments (186 “Drone” segments and 41 “No drone” segments) of RF signal-strength data obtained from three different kinds of drones. These three drones have four different operational modes: on, hovering, flying without video recording, and flying with video recording. These drone operating modes are labeled as Modes 1–4. Information about the Drone RF dataset, including the quantity of segments and raw samples for each kind of drone, is briefly summarized in Table 3 [39]. Additionally, Table 4 demonstrates the detection and identification cases for the open-source dataset [39]. Three machine learning models were created to detect and identify drones using the XGBoost algorithm under the three scenarios: drone presence, drone presence and type, as well as drone presence, type and operating mode. According to the experimental results, in comparison to the other two scenarios, the accuracy of the classifier decreases when utilizing an RF signature to identify and detect the operating modes of drones. This implies that utilizing a signal’s frequency components as a signature to identify drone activity has limited efficacy.

*Drone detection and identification (DDI) systems using Dl algorithms*: The authors of [41] trained a deep neural network (DNN) with four fully connected layers and confirmed the dataset’s validity using 10-fold cross-validation. By contrast, in [42], the authors attempted to tackle the drone detection issue and achieve better accuracy by using a simple CNN rather than DNN. The research in [43] established a new technique for RF-signal-based UAV detection and identification using transfer learning. Firstly, a sparse RF signal was sampled using compressed sensing, and then, in the preprocessing part, a wavelet transform was used to extract time–frequency features of non-stationary RF sample signals. Three-level hierarchical classification was performed using a Visual Geometry Group (VGG)-16 network retrained on the preprocessed compressively sensed RF signals to detect and identify UAV presence, model, and flight mode. UAV detection, type, and operating mode identification capabilities for various compression rates are assessed using the publicly available dataset DroneRF [39]. The experimental findings demonstrated that the proposed technique performs well and has a greater detection rate and recognition accuracy than other methods [40,41,42]. In [44], the authors suggested a compressively sensed RF-signal-based multi-stage deep learning approach for UAV detection and identification. The authors replaced the traditional data sampling theorem with the compressive sensing theory to sample and preprocess the initial data. Then, two neural network models were designed to detect the presence of UAVs as well as to classify the UAV model and flight modes. A DNN network containing five dense layers was used for UAV presence detection. The CNN network consisted of six 1D convolutional layers for feature extraction followed by pooling layers for reducing data size and two fully connected layers for classifying the output data; the network was designed to identify and classify UAV types and flight modes. Different experiments were conducted using the open-access DroneRF dataset [39]. Experimental results were compared to those of other research works [37,38,40,41,42] carried out on the same dataset [39]. By comparing all of these methods, the authors of the paper [45] proved the efficiency of the compressed sensing technique for sampling the communication signals between the controller and the UAV. Furthermore, the experiment outcomes demonstrated that even at extremely low sample rates, the approach employed in their work performs better than alternative learning strategies in terms of evaluation metrics such as accuracy, F1-score, and recall. Another study uses a multi-channel 1-dimensional convolutional neural network (1D-CNN) for drone presence detection and drone type and flight state identification [45]. The authors suggested channelizing the spectrum into several channels and using each channel as a different input for the classifier. Therefore, the whole Wi-Fi frequency spectrum is separated into eight distinct channels, whereby each channel provides unique information about the drone’s RF activity and flight mode. The proposed multi-channel model consists of channelized input data, a feature extractor with two stages of convolutional and pooling layers, as well as a fully connected layer, or MLP, for classification tasks. The experiment was conducted on the DroneRF dataset [39], which features drones operating in various modes. Experimental outcomes demonstrated that the proposed multi-channel 1DCNN performs noticeably better than the methods described in [41]. Related works on DDI using ML and DL methods were briefly compared (see Table 5). A number of deep learning and machine learning models were tested by the authors of [46] for drone RF feature extraction and classification tasks. The proposed drone detection solution started with preprocessing raw data of drone RF signals from the publicly available DroneRF dataset [39]. Then, feature extraction methods such as root mean square (RMS) and zero crossing rate (ZCR) for extracting time domain features; discrete Fourier transform (DFT), power spectral density (PSD), and mel-frequency cepstral coefficients (MFCCs) for extracting frequency domain features; as well as fused methods were used to extract time and frequency domain features. Several drone detection solutions were performed during the experiment. In the first solution, extracted features were fed into an XGBoost machine learning classifier, whereas in the second solution, both feature extraction and classification were performed using a 1D-CNN deep learning model; in the third solution, features were extracted using a 1D-CNN deep learning model and classified using machine learning classifiers and vice versa. K-fold cross-validation was used to compare the performance of each solution. The authors solved the class imbalance problem by using the Synthetic Minority Oversampling Technique (SMOTE) data augmentation method, whereby synthetic data points are generated based on the original points for the minority class. This helped to improve the classification performance of the proposed drone detection approach. According to the experimental results, PSD features with the XGBoost classifier showed better results than other solutions. In [47], the authors proposed a deep learning model based on time–frequency multiscale convolutional neural networks (TFMS-CNNs) for drone detection and identification. The proposed model is made up of two parallel networks with a layered design: one of the networks takes input in the frequency domain and the other one in the time domain. In the preprocessing stage, the raw RF segments were transformed into frequency domain signals using the discrete Fourier transform. The two parallel networks consist of 1D convolutional blocks and global max pooling layers. Experimental results showed that the proposed TFMS-CNN approach, which incorporates both time and frequency domain data, significantly outperforms previous works’ models [41,42,43] that were trained solely on frequency domain signals in terms of accuracy and F1-scores. Table 5 below summarizes the comparison of these ML- and DL-based drone detection and identification methods based on the DroneRF [39] dataset.

### 2.3. Acoustic-Based Detection

*Principles of acoustic-based detection*: Audio-based methods have been shown to be promising for drone identification in recent years. Due to their engines, propeller blades, aerodynamic features, etc., flying drones generate a variety of distinct acoustic signatures, which can be used to aid in detection purposes. Nevertheless, the sound produced by propeller blades is frequently employed for detection because it has a comparatively larger amplitude. Numerous research works have examined the sound produced by drones, using characteristics like frequency, amplitude, modulation, and duration to identify a drone’s existence [48,49,50,51,52,53,54,55,56,57,58,59,60].

Acoustic-based drone detection relies on the use of specialized, highly sensitive audio sensors like microphones or microphone arrays to capture the noises made by propeller blades of drones, and the resulting audio signals are analyzed based on different methods such as correlation/autocorrelation or machine learning to identify the presence, type, and capabilities of drones [48]. Usually, the term “drone detection” refers to the identification of unauthorized drone activity, such as entering a no-fly zone or filming something on camera. However, the expanded process of intruder drone detection often entails figuring out its presence, type, model, and other capabilities such as size, speed, altitude, direction, position, etc. Finding out whether a drone is in the region of interest refers to “drone presence detection” or just the “drone detection” problem. However, determining the precise model or type of the spotted drone is the drone identification problem. The exact geographic position of the identified drone is ascertained using drone localization. Drone fingerprinting is the procedure of processing a captured audio signal and mapping it to those of UAV IDs stored in a database. Drone presence detection performs a binary classification task. Drone identification can be represented as binary classification when determining if a drone is authorized or not, or as multi-class classification by identifying the detected drone by its model, activity, size, unique ID, etc. [17].

*Drone detection and identification using ML algorithms*: The acoustic-based drone detection problem was considered in [49], wherein the authors proposed an inexpensive and robust machine-learning-based drone detection system. Two online databases including different drones’ augmented audio signals were used for drone data preparation. The “No Drone” database is made up of various ambient noises that were gathered from sound databases on YouTube and from the BBC. A total of 26 mid-level audio features, called ‘mel-frequency cepstral coefficients’ (MFCCs) were extracted from both drone and non-drone data. Different duplicate values and outliers were removed in the preprocessing stage. As MFCC features represent high-dimensional drone spectral envelopes, principal component analysis (PCA) was applied to reduce the high-dimensional attributes. K-medoid clustering was applied on the normalized PCA to obtain the ground truth for the total number of drones in the drone database. Two ML classifiers, balanced random forest (BRF) and multi-layer perceptron (MLP), performed binary classification by identifying real-time drone presence in the target area. Experimental results demonstrated that for the test dataset, MLP, with an F1-score of 0.83, gave better results than BRF (the F1-score for BRF was 0.75). The distance dependence of an acoustic-based UAV detection system based on various machine learning algorithms was studied in [50]. The dataset consisted of sounds from several drone models (DJI Phantom 3, Parrot AR, Cheerson CX 10, and Hobbyking FPV250) as well as motorbikes, helicopters, airplanes, and building sites; the labels were “drone” and “no drone”. Although there is no data for frequencies higher than 8 kHz, the signals have been resampled with a sampling frequency of 16 kHz from their original 44.1 kHz sampling frequency. In the feature extraction stage, several spectral features in the frequency domain, like MFCC, delta MFCC, delta-delta MFCC, pitch, centroid, etc., were calculated for every frame. The extracted features were fed to ML classifiers, such as the least squares linear classifier, MLP, radial basis function network (RBFN), support vector machine (SVM), and RF, and these classifiers were compared using distance-based error probabilities. A more-thorough evaluation of the models was provided by using receiver operating characteristic (ROC) curves, for which RF seemed to be the best detector in terms of probability of detection. On the other hand, simple linear classifiers outperformed in terms of distance-based performance variation. Subsequently, the authors suggested combining these two algorithms to achieve the best results. Another similar work was conducted in [51], wherein the authors focused on finding the best parametric representation for ML-based audio drone identification. The drone database used consisted of two commercially available drones’ (Bebop and Mambo) propeller noise recordings for both “Drone” and “No Drone” classes. In the feature extraction part, five acoustic features—MFCC, Gammatone cepstral coefficients (GTCC), linear prediction coefficients (LPC), zero-crossing rate (ZCR), and spectral roll-off (SRO)—were extracted to examine optimal audio descriptors for the drone detection problem. In the classification part, an SVM classifier was employed to classify drone audio data. Experimental findings proved that GTCC can adequately describe the acoustic characteristics of a drone and produce the best outcomes compared to the other feature sets; furthermore, the MFCC and LPC feature sets are also useful for audio drone detection, as demonstrated by their considerable detection performance; on the other hand, ZCR and SRO, as single-value features, perform the worst based on sole utilization. Nevertheless, combining all five audio features yields the best classification outcomes. In [52], the authors presented an ML framework for amateur drone (ADr) identification among other noises in a loud environment. Four distinct sound datasets featuring birds, drones, thunderstorms, and airplanes were gathered for the multi-class classification task. Linear predictive cepstral coefficients (LPCC) and MFCC features were extracted from acoustic sounds. Following feature extraction, these sounds were accurately classified using support vector machines (SVMs) with different kernels. The experimental findings confirmed that the SVM cubic kernel with MFCC performs better than the LPCC approach by reaching an accuracy of about 96.7% for ADr identification. In future works, the authors plan to improve detection accuracy by utilizing deep neural networks with a huge amount of acoustic data. A robust ML-based drone detection framework for acoustic surveillance was established in [53]. The dataset was captured in various sound environments and included various UAV types for training and unseen test sets, as well as sounds of ambient noises such as birds, wind, engines, and building noise. Different block-wise temporal and spectral domain features were extracted to distinguish UAV sounds from background noises. The extracted relevant features were fed into an SVM classifier that enabled the researchers to recognize and classify sounds based on their acoustic profiles. The experimental results were evaluated with a confusion matrix first, whereby robust detection findings were produced with a very low percentage of false positives. Based on other evaluation metrics such as accuracy, F1-score, sensitivity, etc., the proposed SVM-based technique outperformed spectrogram-CNN on the testing dataset. Additionally, the authors checked the robustness of the proposed approach by dividing the binary classes into several subclasses, including specific types of noise disruptions and problematic sound occurrences. The result was a decrease in performance in identifying recordings from more remote or silent UAVs. Also, bird calls and building sounds from the noise category frequently caused confusion, leading to false-positive UAV detection. The authors claimed that these limitations can be fixed by extending training audio recordings, combining audio–visual data into the detection process, or integrating many acoustic sensor outputs into array processing systems. Similar research on acoustic-based UAV recognition based on analyzing UAV rotor sounds was presented in [54]. Hexacopters and quadcopters were employed for creating drone and non-drone datasets from scratch: gradually starting from 21 samples and reaching 696 samples. The samples include several scenarios, such as UAVs landing, taking off, hovering, and doing flyby passes. Feature extraction was performed by MFCCs representing the short-term sound power spectrum. The extracted features were trained on an SVM classifier. For the CNN classifier, a spectrogram saved as a 224 × 224 jpg was made for each sample. The validation accuracy of both classifiers was checked for 21, 110, and 284 samples. Experimental results indicated that in the 21-, 110-, and 284-sample tests, SVM was more accurate than CNN. Due to small and imbalanced training data, the 284-sample test results from CNN saw a steep decline, leading to the decision to shift the focus to the SVM model going forward, which for the 696-sample test achieved a data validation accuracy of 92%.

*Drone detection and identification using DL algorithms*: The authors of [55] concentrated on deep learning approaches for drone detection and identification, including convolutional neural networks (CNNs), recurrent neural networks (RNNs), and convolutional recurrent neural networks (CRNNs). The dataset was acquired by recording Bebop and Mambo drone propeller sound samples with a smartphone’s built-in microphone while the drone was hovering and flying in a quiet indoor space. Preprocessing started with reformatting, which converted the audio file type, sample rate, bitrate, channel, etc. Then, segmentation of the formatted audio files was performed to optimize the model’s training for real-time implementation. The authors overlapped real-world background noises with the drone audio as a data augmentation method. For the first experiment—the drone detection problem—the data acquired for the Bebop and Mambo drones were combined into a single entity labeled as “drone”, and all other audio clips were labeled as “not a drone”. The second experiment focused on the drone identification problem, for which a multi-class classification problem was addressed to identify Bebop and Mambo drone models and unknown noises, including other background noises. The results indicated that in terms of all evaluation metric values, CNN outperformed RNN; however, compared to CNN and convolutional recurrent neural network (CRNN) methods, RNN required the least amount of training time; further, CRNN outperformed RNN but showed decreased performance compared to CNN across all evaluation metrics. Considering that the RNN algorithm is designed to work best with sequential data, short audio clips may have led to a decline in its performance. CRNN and CNN showed close results in terms of performance; however, in terms of training time, CRNN was significantly quicker than CNN. In [56], a CNN-based detection system with a STFT feature extraction technique was used to compare drum sounds and tiny fan sounds to a UAV’s hovering acoustic signal. The acoustic signal data from a drum and a tin were recorded in a silent laboratory with no other noise. UAV hovering acoustic data was provided by IEEE SPCUP 2019. In the preprocessing stage, STFT was applied to the noise produced by the UAV to convert one-dimensional data into two-dimensional features. The proposed simple CNN model consists of seven layers. According to the experimental outcomes, the CNN-based model for UAV detection showed a high detection rate. However, the authors suggested that in the future, small and noise-free data should be verified in a more populated outside area. Another CNN-based multi-class UAV sound classification with a large-scale dataset was proposed in [57]. The large-scale UAV dataset introduced contains audio files of 10 distinct UAVs, ranging from toy drones to Class I drones, with each file including a 5-second recording of the flying drone sound. In the feature extraction part, MFCCs were extracted from audio samples in two steps: first, the audio was converted from the Hz to the Mel scale and then the logarithm of the Mel scale; next, logarithmic magnitude and discrete cosine transformations were performed. The MFCC feature is a cepstrum feature formed from the Mel frequencies as a result of the preceding processes. The extracted features were then fed into a simple CNN model to train the drone classification system. The obtained results were evaluated by loss and accuracy plots as well as evaluation metrics. The resulting F1-score values proved that the CNN-based framework coped with the classification task of different UAV models: showing overall test accuracy around 97.7% and test loss around 0.05%. In [58], audio-based UAV detection and identification employing DNN, CNN, LSTM, convolutional long short-term memory (CLSTM), and transformer encoders (TE) was benchmarked with the dataset used in [55]. In addition to the dataset from [55], the authors gathered their own varied-identification audio dataset, which included seven different kinds of sound categories, such as ’drone’, ’drone-membo’, ’drone-bebop’, ’no-UAV’, ’helicopter’, ’aircraft’, and ’drone-hovering’. Mel-frequency cepstral coefficients (MFCC) were calculated for each audio frame for the feature extraction module. The extracted features were fed into deep neural network models for UAV sound detection and UAV identification tasks. A DNN model was built with three hidden layers, and the output layer depended on a softmax activation function. On the other hand, the CNN model with small-sized inputs had three convolutional layers without any subsequent pooling layers and had a hidden layer with a softmax activation function. The LSTM was built using one 128-unit LSTM layer followed by a hidden output layer with the same activation function as previous models. Likewise, CLSTM employed a convolutional layer and an LSTM layer followed by a hidden output layer. Finally, the TE was built with a positional encoder, two attention heads, a 128-unit encoder layer, and a softmax activation mechanism. A comprehensive comparison of existing ML and DL-based drone detection methods based on acoustic signatures was shown in Table 6.

Researchers have compared the detection and identification capabilities of all of these proposed models with the models established in [55], such as RNN, CNN, and CRNN, for the same dataset. Based on the UAV detection experimental findings, the LSTM model showed the best results compared to all the other models. In the UAV identification experiment, all the proposed models showed better identification accuracy than the models in [55]: specifically, the LSTM model outperformed with high accuracy. The authors of [59] proposed an enhanced dataset version of their study conducted in [55] by employing real-like drone audio clips artificially created by generative adversarial networks (GANs). By comparing experiments based on GAN to those without GAN, the authors concluded by suggesting a strategy of employing GANs demonstrates a promising solution to bridge the gap imposed by the scarcity of drone acoustic datasets while simultaneously improving classifier performance in the majority of cases, whereas in cases for which no significant improvement was observed, the classifier performed similarly to when GAN was not used. Real-time UAV sound recognition based on different RNN networks was conducted in [60]. A drone sound dataset was gathered by flying several drone models with and without payloads. In order to differentiate UAV sounds from false negatives, the data of ambient noises were also gathered, including noises of wind, motorbike riding, and canopy whispers. All the collected data were labeled as “Unloaded UAV”, “Loaded UAV”, or “Background noise” and were examined in the time and frequency domains to determine the intervals of the information domains. Once the desired frequency range was selected, special filters were used to filter out any sounds below 16,000 Hz. Thus, mel-scale features were extracted by employing a variety of time and frequency hyperparameters. UAV sound recognition models were trained based on RNNs such as SimpleRNN, LSTM, bidirectional long short-term memory (BiLSTM), and gated recurrent unit (GRU). Experimental results proved the efficacy of using RNNs for UAV sound recognition, as they outperformed CNN models. For comparison between RNN models, the GRU architecture was discovered to be an efficient model for distinguishing loaded and unloaded UAVs as well as background noise with high accuracy.

### 2.4. Vision-Based Detection

*Principles of visual-based detection:* A visual detection system is based on capturing drone visual data, such as images or videos, using camera sensors and then detecting drones in them using computer-vision-based object detection algorithms. The process of capturing visual data from objects is called image acquisition and consists of three main steps: first, energy is reflected from the object of interest; then, it is focused by an optical system; and finally, the amount of energy is measured using a camera sensor (see Figure 5).

Related works on drone object detection based on visual data can be divided into two major categories depending on whether the authors employed a deep learning model or dealt with feature extraction using traditional handcrafted features. However, recent studies on visual-based drone detection and identification have mostly employed learned features or deep learning models [61,62,63,64,65,66,67,68,69,70,71]. Deep learning for object detection with convolutional neural networks (CNNs) has two approaches: one-stage detection and two-stage detection. Due to their high inference speed, one-stage detectors such as the you only look once (YOLO) and single shot detector (SSD) algorithms are extremely quick. These detectors lack a region proposal stage, which is common in two-stage detectors. Two-stage detectors are more accurate, which implies improved localization and identification. This method seeks to provide several bounding boxes for the objects in the image. The most well-known two-stage detectors are the region-based convolutional neural network (R-CNN), fast R-CNN, faster R-CNN, etc., which are shown in Figure 6.

*Object detection with learned features based on one-stage detectors*: In [61], the authors proposed a novel anti-drone YOLOv5s approach to detect low-altitude small-sized UAVs, which is mainly oriented towards border defense applications. Initially, the low-altitude, small-size object detection problem was solved based on the optimization method for feature enhancement. Secondly, as the YOLOv5 base model includes numerous parameters that are unsuitable for use in embedded systems, the authors used ghost modules and depth-wise separable convolution to decrease the number of model parameters. The proposed lightweight detection approach improved the mAP value by 2.2% and the recall value by 1.8%. Another study in [62] focused on drone detection using single-shot detectors such as YOLOv4, YOLOv5, and detection transformer (DETR). In the preprocessing stage, images from the open-access D-Drone dataset were resized, labeled to obtain ground-truth labels, and then converted to YOLO and common objects in context (COCO) formats. Detection models were trained on one-stage detectors, and the results were evaluated based on evaluation metrics. Experimental outcomes showed that YOLOv5 outperformed the other two detectors in terms of average precision (AP) with 99% and an intersection over Union (IoU) of 0.84. The DETR model showed the worst results. The authors in [63] addressed the real-time mini-UAV detection issue in terms of accuracy and speed of detection by using the one-stage YOLOv5 model. The main contributions of the work are the creation of a custom mini-UAV dataset including drones flying in low-visibility circumstances using a Dahua multisensory network pan, tilt, zoom (PTZ) camera and the redesign of the YOLOv5 model for the recognition of extremely tiny flying objects from an aerial perspective based on features learned by deep CNN. As the receptive field size of the baseline YOLOv5 model is insufficient to detect small flying objects, the structure of the original YOLOv5 was redesigned with two improvements: to collect additional texture and contour information from small mini-UAVs, a fourth scale is added to the three scales of YOLOv5 feature maps; and to decrease feature information loss for small mini-UAVs, feature maps from the backbone network are incorporated into the extra fourth scale. Experimental results demonstrated that all evaluation metric values for improved YOLOv5 outperformed the baseline model. Drone and bird detection based on YOLOv4 and YOLOv5 was conducted in [64]. A custom dataset compiled from several online sources consists of 900 images: 664 images for drones and 236 images for birds. By evaluating the experimental results, the authors concluded that YOLOv5 is quicker at object detection and overcomes the problem of detecting tiny drones and birds in real time with high accuracy. Some limitations of the current study include a short dataset, a class imbalance problem, and a failure to account for background noise. An automated image-based drone detection system based on the fine-tuned YOLOv5 method and transfer learning was presented in [65]. The dataset is available online from Kaggle and consists of 1359 drone images acquired from an Earth-to-drone perspective and labeled for binary classification tasks as drone or background classes. The performance of the proposed model was evaluated using the transfer learning pretrained model weights. To highlight the superiority of the proposed approach, the obtained results were compared to other versions of YOLO, such as YOLOv3 and YOLOv4, as well as the two-stage Mask R-CNN model. The comparative findings proved the efficacy and superiority of the suggested framework, especially for distant and tiny drone detection. A comparative study of drone detection based on one-stage detectors such as YOLO and MobileNetSSDv2 was presented in [66]. Two open-source datasets were used for drone classification. Experimental results: the YOLO architecture delivered high performance in terms of precision and accuracy while retaining comparable FPS and memory usage. A fine-tuned YOLOv4-based automated drone and drone-like object detection framework was established in [67]. A dataset of drone and bird images was gathered from different internet sources. The total number of gathered images was 2395, of which 479 images were for bird classes and 1916 images were labeled for drone classes. To evaluate the performance of the proposed model, the trained model was tested on custom videos in which two types of drones were flown at three different altitudes. According to the experimental results, the YOLOv4 model showed appropriate performance with a mAP of 74.36%. Overall, the fine-tuned YOLOv4 was able to overcome issues with speed, accuracy, and model overfitting. The same authors continued this task in [68], wherein the YOLOv5 detector was employed on the same dataset. The main contribution of this study relied on the usage of a data augmentation technique to artificially solve data shortage issues; thus, the total number of images reached 5749. To evaluate the model, evaluation metrics such as mAP, FPS, F1-score, etc., were calculated. The proposed technique surpassed the authors’ previous methodology in [67] with a mAP of 0.904. Due to its lightweight construction, YOLOv5 detected objects faster than YOLOv4.

*Object detection with learned features based on two-stage detectors*: The authors in [69] presented a comprehensive approach for detecting and tracking unmanned aerial vehicles (UAVs) by proposing a visible-light dataset called the ’DUT Anti-UAV’ dataset. The suggested fusion system reliably identifies and tracks UAVs in real-time by combining image processing, object detection, and tracking methods. The authors evaluated state-of-the-art one-stage and two-stage detection methods, including 14 detectors and 8 trackers retrained on their own dataset. The research also underlines the difficulties with camera-based UAV detection, such as differences in appearance caused by various UAV types, weather, and environmental conditions. To overcome these difficulties, the authors suggested using online adaptation approaches, whereby the detection model may be continually modified to accommodate new UAV features. Numerous experiments revealed that, depending on the detection algorithm, the tracking performance gain might vary, and the authors’ fusion approach could greatly boost the tracking efficiency of all the trackers. However, Faster-RCNN with the VGG-16 version was a superior fusion option for the majority of trackers. Drone detection using a two-stage Mask R-CNN detector with two backbones, such as residual network (ResNet)-50 and MobileNet, was presented in [70]. The model was trained on a dataset [71] with 1000 images of drones. The two backbone networks were compared in terms of mean average precision (mAP). Based on the experimental results, Mask R-CNN with ResNet-50 outperformed Mask R-CNN with the MobileNet backbone.

The accuracy of any UAV detection system might be hindered by different interference factors in a real-world scenario. Table 7 lists the main examples of real-life interference factors for each detection technology.

### 2.5. Sensor Fusion and Other Methods for Drone Detection

The review of the four detection methodologies presented in the previous subsections shows that each of the drone detection modalities has its own set of limitations, and a solid anti-drone system may be supplemented by integrating several modalities. Sensor fusion systems can integrate audio and visual features from acoustic and image sensors [77,78,79] or combine radar and visual imaging systems [80,81]; RF and image sensors [82]; radar, RF, and camera sensors [83]; optical camera, audio, and radar sensors [84]; as well as visible, thermal, and audio sensors [85] to enhance drone identification, tracking, and classification. By integrating the capabilities of several sensory modalities, the method increases the robustness and accuracy of detection systems. Sensor fusion techniques such as early fusion and late fusion can be applied to drone detection systems. These approaches define when and how data from various sensors are merged during the detection and identification of drones.

Early fusion (sometimes referred to as ’data-level’ or ’feature-level’ fusion) entails integrating raw data or extracted features from various sensors prior to any additional processing or analysis. The goal is to integrate data as soon as possible, which results in a more-thorough representation of the information (see Figure 7).

Late fusion (sometimes referred to as ’decision-level’ fusion) is a sensor fusion approach in which the decisions or confidence scores indicating the drone’s presence from numerous separate sensors are integrated at a later stage in the processing pipeline (see Figure 8). These individual decisions or confidence scores are then aggregated to form a final decision. Aggregation is critical in late fusion because it uses a variety of sensors to increase the overall accuracy and resilience of the detection system. Voting systems, weighted averages, Bayesian approaches, and machine learning models are some examples of popular aggregation approaches used in late fusion for drone detection.

*Drone detection and identification based on Wi-Fi signal and RF fingerprint*: As we indicated when describing the main threats posed by drones in Section 1.1, to stop illegal activities such as threatening privacy, smuggling, and even terrorism acts, a solid anti-drone system is needed. Generally, most drones employ a radio communication (RC) transmitter and receiver pair to control them; however, these days, Wi-Fi and Bluetooth have made it possible for drone manufacturers to create gadget controllers that operate on smartphones and tablets. If the drone uses wireless communication, then Wi-Fi enables the transmission of massive amounts of data to and from UAVs using Wi-Fi-related protocols such as 802.11a, 802.11b, 802.11g, 802.11n, and 802.11 within a set control radius. These tiny UAVs have a larger potential for imposing hazards across several sectors of society than drones that just use RC controllers with no Wi-Fi- or Bluetooth-linked mobile applications. Therefore, drone detection and identification using RF signals can be performed based on RF fingerprinting and Wi-Fi fingerprinting approaches. In both approaches, the RF communication signal between the drone and its controller is captured by employing an RF sensing device [73]. RF fingerprinting techniques extract physical layer features and signatures from the captured RF communication signal, while in Wi-Fi fingerprinting approaches, medium access control (MAC) and network layer features are extracted. Multi-stage drone detection and classification using RF fingerprints in the presence of wireless interference signals was conducted in [73]. In the first stage, the captured RF data are preprocessed using wavelet-based multiresolution analysis. The preprocessed RF data are modeled using a two-state Markov-model-based Naïve Bayes detection method to differentiate the RF signal from the noise class. In the second stage, if present, signals from Wi-Fi and Bluetooth sources are recognized based on the detected RF signal’s bandwidth and modulation features. After identifying the input signal as a UAV controller signal, it is classified using several ML algorithms. The best classification accuracy achieved by a k-NN classifier was 98.13% for classifying 15 various controllers based on only three features. The experiment demonstrated that the suggested method can categorize the same make and model of UAV controller without reducing overall accuracy. In [74], the authors developed a mechanism for detecting UAVs with both an RC controller and Wi-Fi- or Bluetooth-linked mobile applications based on radio communication and an internet protocol (IP) address. The detection system was divided into two stages. The first stage involves searching for the presence of any unfamiliar IP address in the range of interest. As the UAV has a Wi-Fi connection, it has its own IP address. The drone detection system employs an algorithm that stores certain IP addresses of known devices in its database. Then, utilizing a Wi-Fi adapter, the algorithm continues to verify whether any unfamiliar device is connected to the router by matching its IP address with those stored in the database. In the event that any unfamiliar device is connected to the router, the system outputs its IP address. The second stage entails determining whether the detected unfamiliar device is a UAV or not by using an RF receiver. Overall, the detection system prototype could be able to detect drone presences over a range of 100 m, even when birds appear in the region of interest.

*Drone detection and identification based on cellular and IoT networks*: With the arrival of 5G and the Internet of Things (IoT), a low-cost bistatic radar system was introduced in [86]. The proposed system uses 5G non-line-of-sight (NLOS) signals bounced off the drone’s body and the drone’s Wi-Fi-received signal strength indicator (RSSI) emission to identify and locate the drone. A k-NN classifier was used to train these two signatures for real-time drone location prediction. The system was designed for indoor and outdoor environments using three HackRF-One SDRs: two HackRF SDRs act as 5G signal transmitters and receivers, while the third HackRF SDR works as a Wi-Fi RSSI receiver. The experimental results showed that the proposed cost-effective system can perform real-time drone detection with zero false negative (FN) outcomes. An ensemble-based IoT-enabled drone detection approach using transfer learning and background subtraction was proposed in [87]. The ensemble model includes two DL models: YOLOv4 and ResNet-50. The proposed detection scheme consists of a Raspberry Pi, cameras equipped with IoT, and ensemble models of DL. A hybrid dataset of birds and drones was collected from publicly available datasets. All the images were preprocessed and sent to the background subtraction module. Then, foreground images were sent to the YOLOv4 and ResNet-50 models. In terms of detection accuracy, the suggested approach outperformed competing existing systems. Wireless communication is based on radio frequency waves, occupies a large transmission spectrum, and has similar hardware components as radar sensing, which leads to the use of 5G [88,89] and beyond-5G (B5G) networks [90] (such as 6G, which is still in the research phase) for detecting, identifying, and managing UAVs. Further, cellular networks have high-speed connectivity and low latency that are critical for the real-time control and monitoring of UAVs. In [88], single- and multi-rotor UAVs were identified by transmitting 5G millimeter waves and employing a joint algorithm such as short-time Fourier transform (STFT) and a Bessel function basis. Drone identification utilizing a passive radar and a fully operating 5G new radio (NR) network as an illumination source was presented in [89]. An intelligent charging–offloading approach for an air–ground integrated network in B5G wireless communication was suggested in [90]. The proposed approach used the UAV as a mobile edge computing (MEC) server and mobile power supply in order to extend the sensor network’s lifetime and improve system performance.

## 3. Discussion and Conclusions

A survey of current drone detection and classification techniques indicates a fast-expanding area characterized by considerable technological advances and novel approaches. The fast development in the usage of UAVs has prompted serious issues about privacy, security, and safety. As a result, developing effective UAV detection algorithms has become critical. The central goal of this review article was to offer a complete overview of present UAV detection and classification approaches, methodologies, and frameworks. Therefore, the paper offered an overview of several UAV detection approaches, such as radar-based, acoustic-based, RF-based, and visual-based approaches. However, the discussion also emphasized the inherent challenges that persist. The size and speed diversity of drones, their dynamic behavior and similarity to other flying objects, and their limited battery life make the detection task more challenging. Additionally, different interference factors in real-world scenarios, including adverse weather and lighting conditions, urban locations with plenty of obstructions such as buildings and trees, ambient and background noise, wind, the presence of wireless communication signals from Wi-Fi and Bluetooth sources, bird echoes, etc., present unique challenges for each detection modality.

From Table 2, it is evident that no single detection method is a panacea; each of the drone detection modalities has its own set of limitations, and a solid anti-drone system may be supplemented by integrating several modalities. Integrating the capabilities of several sensory modalities can increase the robustness and accuracy of a detection system. According to the reviewed research works, early and late fusion can be applied to integrate individual sensor modalities. These approaches define when and how data from various sensors are merged during the detection and identification of drones. In early fusion, raw data or relevant features from various sensors are combined before the model, which is trained by normalizing data ranges or concatenating the feature vectors. The primary benefit of early fusion is that it allows the detection system to fully utilize the combined data, perhaps resulting in more accurate detection and classification results. Early fusion necessitates that all sensor data be consistent in terms of size, resolution, and temporal alignment. Therefore, it might be more computationally intensive, as the fused dataset may be huge and complicated. In late fusion, the decisions or confidence scores indicating a drone’s presence from individual sensors are combined at the detection phase. The independent decisions are then aggregated using different aggregation approaches.

These days, most drone manufacturers produce Wi-Fi- and Bluetooth-enabled UAVs with gadget controllers that operate on smartphones and tablets. These tiny UAVs have a larger potential for imposing hazards in several sectors of society than drones that just use RC controllers with no Wi-Fi- or Bluetooth-linked mobile applications. Therefore, drone detection and identification using RF signals based on Wi-Fi fingerprinting is also an important field. Wi-Fi fingerprinting approaches extract the MAC and network layer features in the feature extraction stage. Regarding this approach, different multistage drone detection methods using Wi-Fi and RF fingerprinting have been proposed by researchers.

With the arrival of 5G networks and the Internet of Things (IoT), drone detection systems have been expanded. Wireless communication is based on radio frequency waves, occupies large transmission spectra, and has hardware components similar to those used for radar sensing. Therefore, these properties of cellular networks lead their use as bistatic radar systems for effective drone detection.

This review article seeks to offer a complete overview of the state-of-the-art approaches, methodologies, and challenges in the field of UAV detection and classification, paving the way for developments that will answer the rising concerns about UAV operations. We hope that this review article will be a helpful resource for academics, engineers, and policymakers working in UAV detection and classification since it consolidates the information and insights gathered from a variety of research activities. It might also throw light on future research paths: highlighting prospective avenues to improve the efficacy, efficiency, and dependability of UAV detection systems.

## Figures and Tables

**Figure 1 sensors-24-00125-f001:**
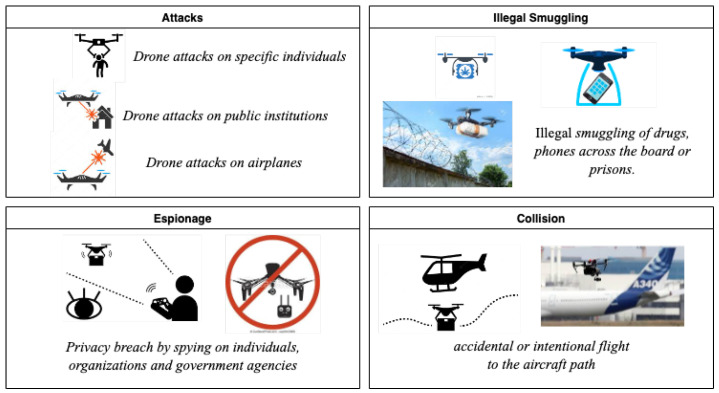
The main drone threat categories: drone attacks, illegal smuggling, drone espionage, and drone collisions.

**Figure 2 sensors-24-00125-f002:**
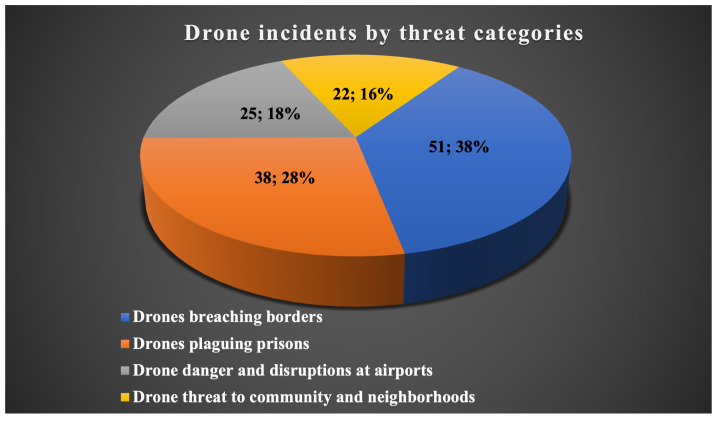
Recorded drone incidents by main threat category.

**Figure 3 sensors-24-00125-f003:**
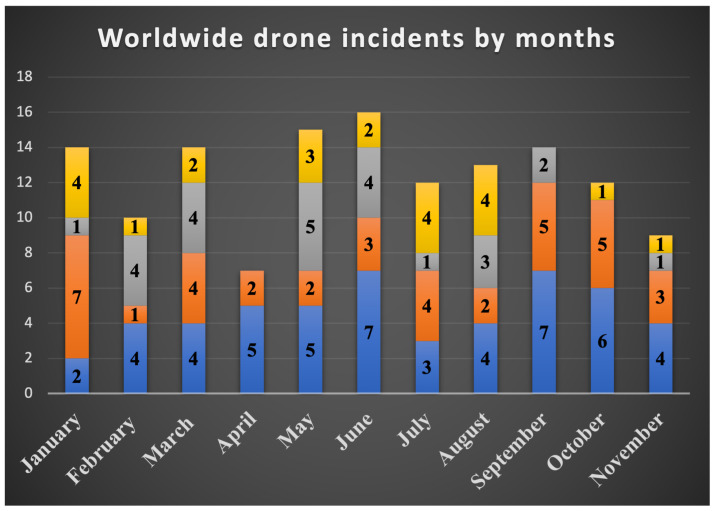
Worldwide drone incidents from January until December 2023. Four types of threats are highlighted in four colors: drones breaching borders in blue, drones plaguing prisons in orange, drone danger and disruptions at airports in gray, and drone threats to communities and neighborhoods in yellow.

**Figure 4 sensors-24-00125-f004:**
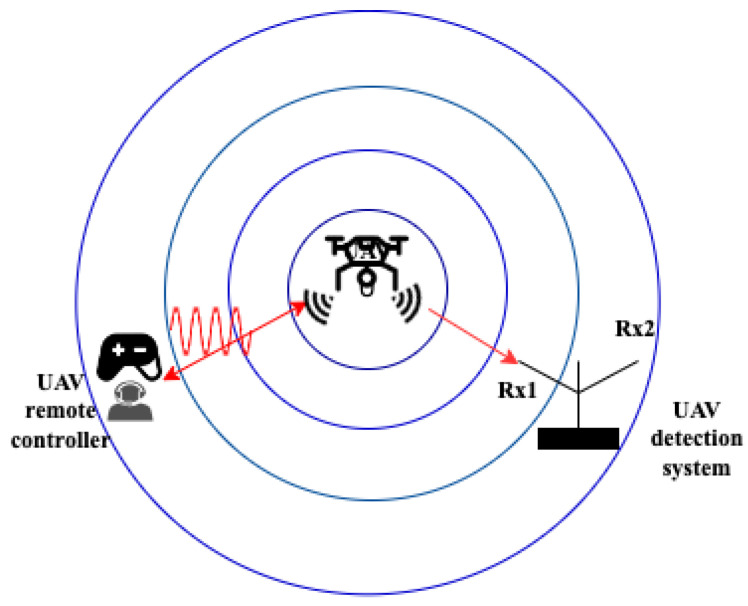
RF-based UAV detection system components: UAV, UAV remote controller, and two receiver units for capturing lower and upper bands of RF signals [37].

**Figure 5 sensors-24-00125-f005:**
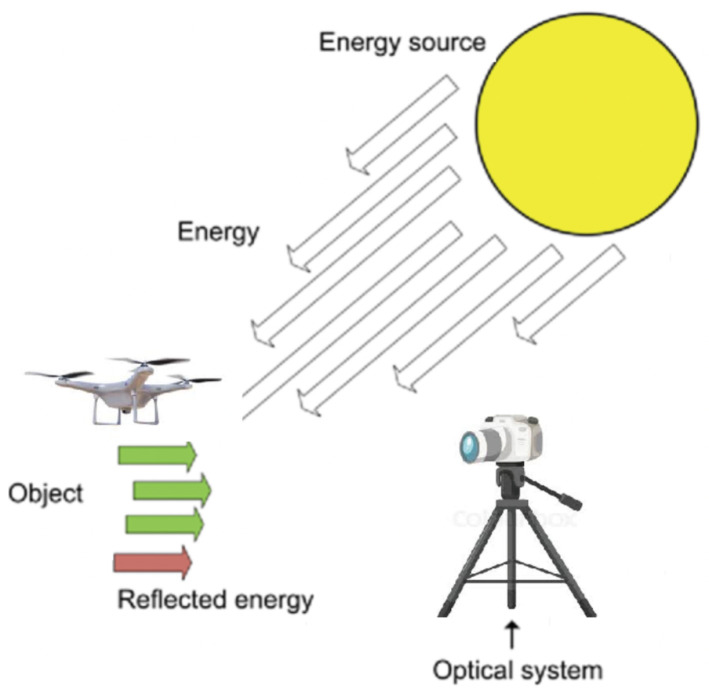
The main steps of the image acquisition process are: energy is reflected from the UAV, the reflected energy is focused by an optical system, and a camera sensor measures the amount of energy.

**Figure 6 sensors-24-00125-f006:**
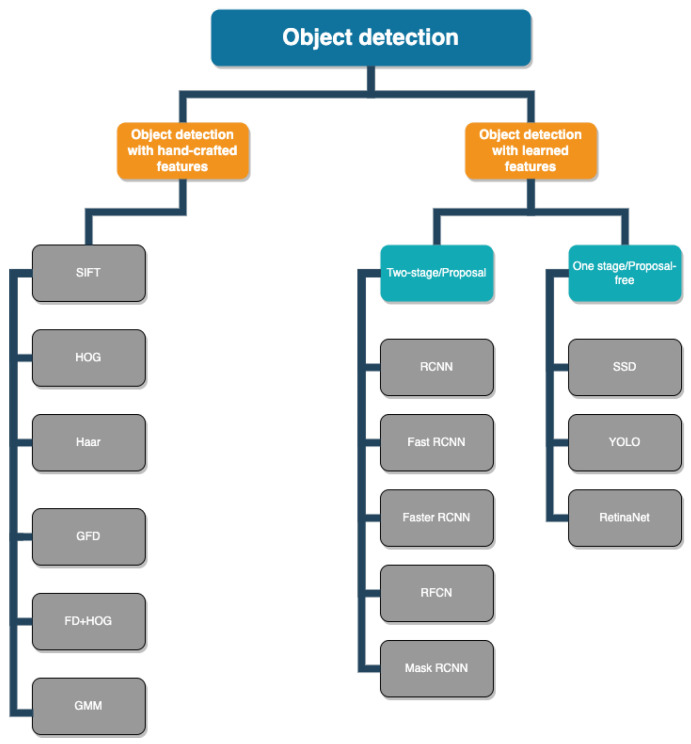
Visual-based object detection methods. Object detection with low-level hand-crafted features use edges, blobs, and color information. Object detection with learned features use extensive DL models.

**Figure 7 sensors-24-00125-f007:**
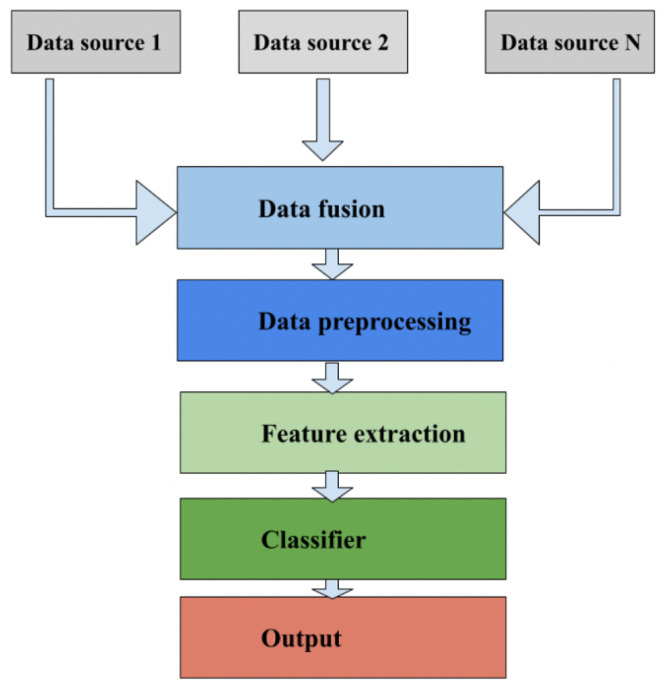
Early sensor fusion. Sensor fusion is accomplished by fusing raw data from individual sensors, and the fused data are fed to a single detection system.

**Figure 8 sensors-24-00125-f008:**
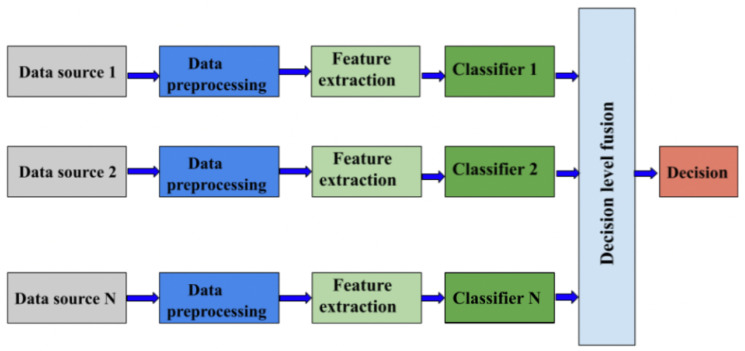
General scheme for late fusion. Each sensor modality is trained separately to detect drones. Sensor fusion is achieved by fusing the detection decisions of individual sensor modalities.

**Table 1 sensors-24-00125-t001:** The list of recent drone incidents reported in the press worldwide.

Date and Location	Type of Threat Category	Incident Details	Response
13 August 2023, Absecon, NJ, USA	drone attack	a business owner of a heating and air conditioning company accused of using a drone to drop harmful chemicals into commercial and residential pools [7]	drone spotted by authorities
3 June 2022, American Canyon, CA, USA	drone attack	a 55-year-old man was detained after using his drone to discharge lit illegal M-80-style explosive devices [8]	drone intercepted by authorities
11 July 2023, County Durham, UK	drone smuggling	a 56-year-old man was accused of using a drone to fly contraband items into a Stockton’s Holme house prison [9]	a police dog spotted and caught the man
16–19 June 2023, Kingston, Canada	drone smuggling	over a kilogram of cannabis and other unauthorized items were confiscated at Collins Bay Institution [10].	the Correctional Service of Canada seized the flying drone
22 June 2023, Canterbury, New Zealand	drone espionage	a lifestyle block owner shot down a drone above his property because he thought it was being controlled by a thief [11]	the incident was not reported to the police
25 May 2023, Lebanon border	drone espionage and surveillance	a DJI quadcopter flying over the border from Lebanon was shot down [12]	Israeli forces shot down the drone using electronic warfare techniques
2 December 2022, Stansted Airport, UK	near collision	a holiday plane carrying up to 189 passengers had a lucky escape from an illegally flown drone that was only a few feet away [13]	the incident was reported by the UK Airprox Board
14 May 2023, Gatwick Airport, UK	near collision	all runways were closed due to a drone sighting [14]	the incident was reported to the police and the proper aviation authorities
22 May 2023, Gao International Airport, Mali	drone collision	all the runways were closed due to a drone crash [15]	the incident was brought under control by the airport staff and security forces
22 May 2023, Katowice Airport, Poland	drone collision	a low-cost Wizz Air airliner flew dangerously close to a huge quadcopter with only 50 m of clearance [16].	the incident was reported to the police and the proper aviation authorities

**Table 2 sensors-24-00125-t002:** Comparison of different drone detection technologies.

Detection Technique	Principle of Operation	Advantages	Disadvantages
Radar-based	employs radio waves to detect and locate nearby objects	long range; all-weather performance; ability to recognize micro-Doppler signatures (MDS); speed and direction measurement	limited detection capability due to low radar cross section (RCS); limited performance due to low altitudes and speeds; high cost and complexity of deployment
RF-based	captures wireless signals to detect the radio frequency signals from drones	long-range detection and identification; resistance to all weather conditions; ability to capture signals and communication spectra from the UAV and its operator; ability to distinguish different types of UAVs	unable to identify autonomous drones; interference with other RF sources; vulnerable to hackers
Acoustic-based	detects drones by their unique sound signatures	cost effective; no line-of-sight (LoS) required; quick deployment	background noise; limited detection range; vulnerability to wind conditions
Vision-based	captures drone visual data using camera sensors	visual confirmation; non-intrusive; cost-effective	limited detection range and requires LoS; weather and lighting dependence

**Table 3 sensors-24-00125-t003:** An overview of the DroneRF dataset [39].

Drone Type	Number of Segments	Number of Samples	Ratio, %
Parrot Bebop	84	$1680\times 10^{6}$	37
Parrot AR	81	$1620\times 10^{6}$	35.68
DJI Phantom 3	21	$420\times 10^{6}$	9.25
No Drone	41	$820\times 10^{6}$	18.06

**Table 4 sensors-24-00125-t004:** Detection and identification cases for DroneRF dataset [39].

Case Name	Class Name	Number of Segments
2-class problem	Drone	186
	No Drone	41
4-class problem	Bebop	84
	AR	81
	Phantom	21
	No drone	41
10-class problem	Bebop mode 1	21
	Bebop mode 2	21
	Bebop mode 3	21
	Bebop mode 4	21
	AR mode 1	21
	AR mode 2	21
	AR mode 3	21
	AR mode 4	18
	Phantom mode 1	21
	No drone	41

**Table 5 sensors-24-00125-t005:** Comprehensive comparison of existing DDI methods based on DroneRF dataset [39].

Preprocessing	Feature Extraction	Classification Method	Accuracy for 10-Class Problem, %	Ref.
Data engineering: a smoothing filter with a window of 15 was applied to each RF signal in order to remove noise and clutter.	Feature engineering: data segmentation using fast Fourier transform (FFT).	Hierarchical approach based on voting principle between two ML algorithms such as XGBoost and KNN	99.2%	[37]
given RF raw segments	Feature extraction in frequency domain using discrete Fourier transform (DFT)	six ML algorithms: XGBoost, AdaBoost, decision tree, random forest, k-nearest neighbor, multilayer perceptron	XGBoost algorithm performs well for 10-class problem: 79.25%	[38]
given RF raw segments	discrete Fourier transform (DFT) of lower band (LB) and upper band (UB) segments	XGBoost	70.09%	[40]
given RF raw segments	convolutional layer extracts features from the data	DNN network with four fully connected layers	not given	[41]
reshape function is applied on the input data	by using filters on the input data, 1D convolutional layer extracts features from the data; average pooling layer followed by conv layer reduces the space dimension of the extracted data	dense or fully connected layer with activation function performs classification	59.20%	[42]
compressive sampling	wavelet transform extracts richer time–frequency information	transfer-learning-based VGG-16	90.2%	[43]
compressive-sensing-based sampling; additionally, ZCC and PSD computation	1D convolutional layers each followed by pooling layers extract the features for CNN network	5 dense layers for DNN network; 2 fully connected layers and activation functions for CNN network perform classification tasks	99.3%	[44]
data channelizing	feature extraction with two stages of convolutional and pooling layers	fully connected layer or MLP for classification tasks	87.4%	[45]
SMOTE data augmentation method for solving imbalanced data problem	RMS and ZCR for extracting time domain features; as well as DFT, PSD, and MFCC for extracting frequency domain features	XGBoost ML and 1D-CNN DL algorithms	99.51%	[46]
DFT transformed the raw FR segments into frequency domain signals	both time and frequency domain features are extracted using two parallel networks	fully connected layers of time–frequency multiscale convolutional neural network	87.67%	[47]

**Table 6 sensors-24-00125-t006:** Comprehensive comparison of existing drone detection methods based on acoustic features.

Dataset	Extracted Features	Classification Method	Experiment Results	Ref.
two online databases	26 mid-level MFCCs	BRF and MLP	F1-score for MLP was 0.83, and for BRF was 0.75	[49]
dataset of several drone models’ audio (DJI Phantom 3, Parrot AR, Cheerson CX 10, Hobbyking FPV250)	MFCC, delta MFCC, delta–delta MFCC, pitch, centroid	least squares linear classifier, MLP, radial basis function network (RBFN), SVM, and RF	ROC curves; RF classifier showed the best detection probability; simple linear classifier outperformed in terms of distance-based performance variation	[50]
drone database consists of Bebop and Mambo drones’ propeller noise recordings	MFCC, GTCC, LPC, ZCR, and SRO	SVM	combination of five audio features yields the best classification outcomes	[51]
sound datasets of birds, drones, thunderstorms, and airplanes	LPCC and MFCC	SVM with different kernels	SVM cubic kernel with MFCC performs better than the LPCC approach by reaching an accuracy of about 96.7%	[52]
sound dataset of various UAV types	different block-wise temporal and spectral domain features	SVM classifier	the experimental results were evaluated with confusion matrix, accuracy, F1-score, sensitivity	[53]
sound data of hexacopters and quadcopters	MFCCs	SVM and CNN classifiers	data validation accuracy for SVM showed 92%	[54]
drone dataset was acquired by recording the Bebop and Mambo drone propellers’ sounds	audio file type, sample rate, bitrate, and channel are converted based on reformatting; then, segmentation was performed	CNN, RNN, and CRNN	CNN outperformed other methods	[55]
dataset of drum sounds and tiny fan sounds plus UAVs hovering	STFT	CNN	CNN-based model for UAV detection showed high detection rate	[56]
large-scale UAV dataset containing audio files of 10 distinct UAVs	MFCCs	simple CNN model	overall test accuracy around 97.7% and test loss around 0.05%	[57]
data sounds of 7 UAV classes	MFCCs	DNN, CNN, LSTM, CLSTM, and TE	LSTM model outperformed with high accuracy	[58]
drone sound dataset was gathered by flying several drone models with and without payloads	MFCCs	SimpleRNN, LSTM, BiLSTM, and GRU	GRU outperformed in distinguishing loaded and unloaded UAVs	[60]

**Table 7 sensors-24-00125-t007:** Real-world interference factors for drone detection technologies.

Detection Technology	Interference Factor	Real-World Scenario	Impact on Detection
Radar	birds and wildlife	bird echoes	reduces the likelihood of distinguishing radar echoes from drones [72];
RF	frequency overlap; signal jamming	presence of wireless communication signals from Wi-Fi and Bluetooth sources [73,74,75]; near military installations or during jamming attacks	interference from other devices operating on similar frequencies; inability to detect RF signals from drones
Acoustic	ambient noise; background noise; wind	noise of high-traffic areas, such as motorbike riding [60]; sounds of birds, airplanes, rain and thunderstorms [76]; open fields or coastal areas	masking of drone’s acoustic signature; distortion of sound waves, making detection difficult
Visual	lighting [21] and weather conditions [17]; obstructions; moving objects [20]	nighttime [21] or varying weather conditions such as fog, rain, and snow [17]; forested areas or cluttered environments; birds, airplanes, insects, and moving parts of scenes [20]	reduced visibility; line-of-sight blockages preventing visual identification; increased false negatives

## Data Availability

Not applicable.

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
