# Peer review of "Advances and Challenges in Drone Detection and Classification Techniques: A State-of-the-Art Review"

_sensors, 2023, doi:10.3390/s24010125_

Round 1

Reviewer 1 Report

Comments and Suggestions for Authors

In this preliminary manuscript, the authors aim to outline both advancements and challenges associated with drone detection and classification techniques. The primary focus encompasses Radar-Based Detection, Radio Frequency (RF)-Based Detection, Acoustic-Based Detection, and Vision-Based Detection.

However, as an invited reviewer, I have several concerns. Firstly, the introduction lacks a compelling justification for the significance of the topic. In Figure 1, depicting the parameters of hazardous UAVs, there is no indication of weight or levels of importance, rendering the simplistic intersection of the four categories somewhat rudimentary.

Section 1.1 lists recent incidents, but lacks statistical support and a global perspective on threats and incidents. In Section 1.2, the authors enumerate Key Challenges in UAV Detection, but these challenges are not thoroughly discussed for each detection technology in subsequent sections. Table 1 is deemed unnecessary; a simple reference with a link to the source website would suffice.

Moving on to the main content (Sections 2.1 to 2.4), where Radar, RF, Acoustic, and Vision-Based Detection methods are discussed, the manuscript falls short of employing a scientific methodology for the review. It fails to concisely present key findings or effectively synthesize and integrate insights from the reviewed literature.

The Discussion and Conclusions section is notably brief and concludes weakly. Additionally, numerous sentences are challenging to comprehend, making it difficult for readers to discern the intended message. For instance, lines 286-288 pose ambiguity with the statement, "Even when using very sensitive radar systems, it is not possible to differentiate between multi-copters and birds with sufficient accuracy due to neither the physical size nor the radar cross-section (RCS)."

In summary, the manuscript requires substantial improvement in justifying the topic's significance, presenting data, discussing challenges thoroughly, eliminating unnecessary elements like Table 1, and enhancing the overall clarity and scientific rigor of the review.

Comments on the Quality of English Language

English very difficult to understand/incomprehensible

Author Response

Dear Dr. Reviewer,

Thank you for giving us the opportunity to submit a revised draft of the manuscript titled “Advances and Challenges in Drone Detection and Classification Techniques: A State-of-the-Art Review” to “MDPI Sensors." We appreciate your precious time and effort in reviewing our manuscript and providing valuable feedback. We are also grateful for your immensely helpful suggestions and insightful comments that led to possible improvements in the current version. The comments are reviewed carefully, and the point-by-point response to each individual comment is provided in the following:

All modifications in the manuscript have been highlighted in blue.

INTRODUCTION

Comment 1: [The introduction lacks a compelling justification for the significance of the topic].

Response: Agreed. Thank you for pointing this suggestion out. Accordingly, throughout the manuscript, we have revised the introduction section by adding the suggested content (please see lines between):

The central goal of the review paper: lines 47–53

The key objectives of the review paper: lines 56–62

Significance of the research: lines 82–84

Research problem: lines 126–129.

Research motivation: lines 81–82.

Comment 2: [In Figure 1, depicting the parameters of hazardous UAVs, there is no indication of weight or levels of importance, rendering the simplistic intersection of the four categories somewhat rudimentary].

Response: We think this is an excellent suggestion. We have incorporated your comment and removed Figure 1 from the manuscript completely. As well, we have made some changes to the sentences.

Comment 3: [Section 1.1 lists recent incidents but lacks statistical support and a global perspective on threats and incidents].

1) statistical support on threats and incidents:

Response: We think this is an excellent suggestion. We have incorporated your comment and condensed the text content about drone threats and incidents by indicating the statistical support for recent drone incidents in Table 1.

2) A global perspective on threats and incidents:

We have added the following content regarding your comment (lines 118-124):

The countries with the highest number of drone occurrences based on analysis of recent drone incidents reported by the press worldwide are: India, the USA, the UK, and Canada.  Most of the drone incidents related to illegal items being delivered into prisons using drones, drone flights over airports, and near borders or restricted zones. The majority drone models involved in the recent drone incidents are DJI Mavic 2, DJI Phantom 4, DJI Mavic 3, etc.

Comment 4: [In Section 1.2, the authors enumerate Key Challenges in UAV Detection, but these challenges are not thoroughly discussed for each detection technology in subsequent sections].

Response: Thank you for this suggestion. We have added suggested content for each drone detection technology, respectively.

1) In Table 2, “Comparison of different drone detection technologies," we have listed the disadvantages of each drone detection technology.

2) In Table 7, “The real-world interference factors in drone detection technologies," we have listed the interference factors in real-world scenarios for each detection technology, respectively.

Comment 5: Table 1 is deemed unnecessary; a simple reference with a link to the source website would suffice.

Response: Thank you for your kind suggestion. According to your comment, we have completely removed Table 1 from the manuscript.  The information about radar frequency bands was mentioned by indicating the appropriate reference in lines 217-220 (therefore, the sentence in lines 209–211 was modified and moved to 217-220).

SECTION 2

Comment 1: Moving on to the main content (Sections 2.1 to 2.4), where radar, RF, acoustic, and vision-based detection methods are discussed, the manuscript falls short of employing a scientific methodology for the review. It fails to concisely present key findings or effectively synthesize and integrate insights from the reviewed literature.

Response: Thank you for your valuable feedback. We did our best to modify the main section of the manuscript based on suggested comments.

1) Firstly, we have removed the text format of advantages and limitations for each drone detection technology, and instead we have made a comparison table of drone detection technologies (see briefly discussed advantages and disadvantages of each drone detection technology in Table 2).

2) Secondly,we have analyzed Ml and DL-based drone sound detection methods by adding Table 6: “Comprehensive comparison of existing drone detection methods based on acoustic features."

3) Thirdly, we have added Table 7 with the name “Real-world interference factors in drone detection technologies” by indicating the main examples of real-life interferences for each detection technology, respectively.

4) Accordingly, throughout the manuscript, we have revised Section 2 by adding the extra subsection 2.5 containing suggested content about sensor fusion technologies, drone detection and identification based on Wi-Fi wireless communication, as well as cellular and IoT networks.

DISCUSSION AND CONCLUSIONS

Comment 1: [The Discussion and Conclusions section is notably brief and concludes weakly].

Response: Thank you very much for the comment. We did our best to correct these mistakes by rewriting the “Discussion and Conclusions” section.

Comment 2: Additionally, numerous sentences are challenging to comprehend, making it difficult for readers to discern the intended message. For instance, lines 286-288 pose ambiguity with the statement, "Even when using very sensitive radar systems, it is not possible to differentiate between multi-copters and birds with sufficient accuracy due to neither the physical size nor the radar cross-section (RCS)."

Response: Thanks for your kind reminders. Indeed, most sentences in the “Discussion and Conclusions” section are not easy to understand. Therefore, we have rewritten this section as well as the Abstract section.

P/S: Dear Reviewer, first of all, we would like to thank you for your extra valuable time, attention, and kindly response to our research and the corresponding manuscript. We do thank you very much for your detailed feedback, suggestions, and comments on our manuscript.  Indeed, your comments were immensely valuable and helpful.  We hope the manuscript, after careful revisions, meets your high standards. If not, we apologize in advance. In this regard, we would also like to mention the problem of lack of time, and we intend to use some of your recommendations in future works. Thank you very much.

Sincerely, authors

Reviewer 2 Report

Comments and Suggestions for Authors

This review paper consolidates information and insights from various research activities, providing a valuable resource for academics, engineers, and policymakers working in UAV detection and classification. It sheds light on prospective avenues to enhance the efficacy, efficiency, and dependability of UAV detection systems while addressing the rising concerns associated with UAV operations. Generally, this paper is easy to follow, but I have some concerns that should be considered in the revised version.

1. The abstract of this review paper lacks specific details and does not provide a clear summary of the main findings or contributions of the paper. It would be helpful to include specific examples of the recent advancements in UAV detection and identification, as well as any significant limitations or challenges that were discovered. 

2. The introduction should present a concise and focused discussion of the main topic, which in this case is Drone Detection and Classification. It should clearly state the objectives and purpose of the paper, explaining why this review is important and what knowledge gap or question it aims to address. Specific incidents involving drones or the different types of threats they pose can be condensed or moved to later sections of the paper where they are more relevant.

3. In Section 2, the author presents a review from four aspects. However, it is important to also consider some integrated methods that merit attention.

4. The Conclusion and Discussion section should contain a clear and concise statement addressing the shortcomings or gaps in existing UAV detection and classification approaches. Moreover, it is necessary to add suggestions and recommendations for future research and development in the field to ensure a more comprehensive conclusion.

5. The figure captions should be enhanced with more detailed descriptions. Including specific information about the content of each figure is crucial as it enables readers to better understand the significance and purpose of the figures.

Comments on the Quality of English Language

Although there are only a few grammar errors, the consistency and conciseness of the expression still need to be improved.

Author Response

Dear Dr. Reviewer,

Thank you for giving us the opportunity to submit a revised draft of the manuscript titled “Advances and Challenges in Drone Detection and Classification Techniques: A State-of-the-Art Review” to “MDPI Sensors." We appreciate your precious time and effort in reviewing our manuscript and providing valuable feedback. We are also grateful for your immensely helpful suggestions and insightful comments that led to possible improvements in the current version. We have carefully considered the comments and tried our best to address each of them.

Below, we provide a point-by-point response to the comments and concerns. All modifications in the manuscript have been highlighted in blue.

Comment 1: [The abstract of this review paper lacks specific details and does not provide a clear summary of the main findings or contributions of the paper. It would be helpful to include specific examples of the recent advancements in UAV detection and identification, as well as any significant limitations or challenges that were discovered]. 

Response: This is a good point! Thank you very much for the comment. We did our best to correct these mistakes by rewriting the “Abstract” section.

Comment 2: [The introduction should present a concise and focused discussion of the main topic, which in this case is Drone Detection and Classification. It should clearly state the objectives and purpose of the paper, explaining why this review is important and what knowledge gap or question it aims to address.]

Response: Agreed. Thank you for pointing this suggestion out. Accordingly, throughout the manuscript, we have revised the Introduction section by adding the suggested content (please see lines between):

The central goal of the review paper: lines 47–53

The key objectives of the review paper: lines 56–62

Significance of the research: lines 82–84

Research problem: lines 126–129.

Research motivation: lines 81–82.

Comment 3: [Specific incidents involving drones or the different types of threats they pose can be condensed or moved to later sections of the paper where they are more relevant.]

Response: We think this is an excellent suggestion. We have incorporated your comment and condensed the text content about drone threats and incidents by indicating the statistical support for recent drone incidents in Table 1.

Comment 4: [In Section 2, the author presents a review from four aspects. However, it is important to also consider some integrated methods that merit attention].

Response: Thank you very much for your valuable feedback! Accordingly, throughout the manuscript, we have revised Section 2 by adding the extra subsection 2.5 containing suggested content about sensor fusion technologies, drone detection and identification based on Wi-Fi wireless communication, as well as cellular and IoT networks.

Comment 5: [The Conclusion and Discussion section should contain a clear and concise statement addressing the shortcomings or gaps in existing UAV detection and classification approaches. Moreover, it is necessary to add suggestions and recommendations for future research and development in the field to ensure a more comprehensive conclusion].

Response: Thank you very much for the comment. We did our best to correct these mistakes by rewriting the “Discussion and Conclusions” section.

Comment 6: [The figure captions should be enhanced with more detailed descriptions. Including specific information about the content of each figure is crucial as it enables readers to better understand the significance and purpose of the figures.]

Response: Thanks for your comment. We agree with you and have revised all of the figure captions with specific descriptions to make them clearer.

P/S: Dear Reviewer, first of all, we would like to thank you for your extra valuable time, attention, and kindly response to our research and the corresponding manuscript. We do thank you very much for your detailed feedback, suggestions, and comments on our manuscript.  Indeed, your comments were immensely valuable and helpful.  We hope the manuscript, after careful revisions, meets your high standards. If not, we apologize in advance. The authors welcome further constructive comments, if any. Thank you very much.

Sincerely, authors

Reviewer 3 Report

Comments and Suggestions for Authors

This review paper highlighted the recent advancements in UAV detection and identification through an in-depth examination of the current literature, including the development of intelligent sensors, machine learning techniques, and deep learning-based architectures. The discussed issue shows the research significance, and the structure and writing are relative good. Only minor revisions are suggested before the acceptance.

1. What is the supposed underlying network communication protocol in this work.

2. The performance discussion on the battery life in this work can be added.

3. The acronyms (eg., RF) should be explained when they first appear, even in Abstract.

4. It is better to resize the oversized figures.

5. The interference factors in real-world scenarios can be discussed.

6. Go through the paper carefully, and remove the typos and formatting issues (e.g., “3) Pulse Doppler Radar:” -> “3) Pulse doppler radar:”?).

7. Make the References more comprehensive, besides this work, some other promising scenarios (e.g., Big data, other IoT systems) can be covered in this work. If the above related work can be discussed, it can strongly improve the research significance. For the improvement, the following papers can be considered to make the references more comprehensive.

Jin Wang, Caiyan Jin, Qiang Tang, Naixue Xiong, Gautam Srivastava, Intelligent Ubiquitous Network Accessibility for Wireless-Powered MEC in UAV-Assisted B5G, IEEE Transactions on Network Science and Engineering, vol.8, no.4, pp.2801-2813, 2021.

Comments on the Quality of English Language

The quality of English writing is good enough.

Author Response

Dear Dr. Reviewer,

Thank you for giving us the opportunity to submit a revised draft of the manuscript titled “Advances and Challenges in Drone Detection and Classification Techniques: A State-of-the-Art Review” to “MDPI Sensors." We appreciate your precious time and effort in reviewing our manuscript and providing valuable feedback. We are also grateful for your immensely helpful suggestions and insightful comments that led to possible improvements in the current version. We have carefully considered the comments and tried our best to address each of them.

Below, we provide a point-by-point response to the comments and concerns. All modifications in the manuscript have been highlighted in blue.

Comment 1: [What is the supposed underlying network communication protocol in this work].

Response: Thank you for your valuable feedback. We did our best to consider your comment in the revised manuscript. Generally, drone remote control systems need robust and efficient communication protocols to guarantee that drones may be securely and effectively managed while also ensuring steady and secure communication between the drone and its operator. The main communication protocols for UAV remote control are RF (ISM band), Digital Enhanced Cordless Telecommunications (DECT) (1.9 Hz), Wi-Fi (2.4-5.8Hz), cellular networks such as 5G and Long Range (LoRa), as well as innovative Light Fidelity (Li-Fi) protocols, etc. The RF protocol enabling data transmission between the drone and its controller is considered one of the most widely used communication protocols for drone remote controllers. The most frequent drone communication frequencies are 2.4 GHz and 5.8 GHz, both of which are part of the Industrial, Scientific, and Medical (ISM) radio bands, which were already mentioned in our manuscript.  These frequencies were chosen for their potential to provide a balance of range and data speed, as well as compatibility with existing Wi-Fi and Bluetooth technologies. The majority of drone models considered in the manuscript are the DJI Phantom, Parrot, and Mavic, which use three forms of transmission systems such as Wi-Fi, OcuSync, and Lightbridge [87].

 In the subsection “Drone detection and identification based on Wi-Fi signal and RF fingerprint,"  we have listed the underlying network communication protocols for data transmission of Wi-Fi-enabled drones.

Drone detection and identification based on Wi-Fi signal and RF fingerprint

As we indicated the main threats posed by drones in Section 1.1, to stop illegal activities such as threatening privacy, smuggling, and even terrorism acts, a solid anti-drone system is needed. Generally, most drones employ a radio communication (RC) transmitter and receiver pair to control them; however, these days, Wi-Fi and Bluetooth have made it possible for drone manufacturers to create gadget controllers that operate on smartphones and tablets.  If the drone uses wireless communication, then Wi-Fi enables the transmission of massive amounts of data to and from UAVs using Wi-Fi-related protocols such as 802.11a, 802.11b, 802.11g, 802.11n, and 802.11 within a set control radius. These tiny UAVs have a larger potential for imposing hazards in several sectors of society than drones that just use RC controllers with no Wi-Fi or Bluetooth-linked mobile applications. Therefore, drone detection and identification using RF signals can be performed based on RF fingerprinting and Wi-Fi fingerprinting approaches. In both approaches, the RF communication signal between the drone and its controller is captured by employing an RF sensing device [85]. To detect and identify the UAVs, RF fingerprinting techniques extract physical layer features and signatures from the captured RF communication signal, whereas in Wi-Fi fingerprinting approaches, medium access control (MAC) and network layer features are extracted. Multistage drone detection and classification using RF fingerprints in the presence of wireless interference signals from Wi-Fi and Bluetooth sources was conducted in [85]. In the first stage, the captured RF data is preprocessed using wavelet-based multiresolution analysis. Then preprocessed RF data is modeled using a two-state Markov model-based Naïve Bayes detection method to differentiate the RF signal from the noise class. In the second stage, if present, signals from Wi-Fi and Bluetooth sources were recognized based on the detected RF signal’s bandwidth and modulation features.  After identifying the input signal as a UAV controller signal, it is classified using several ML algorithms. The best classification accuracy achieved by the kNN classifier was 98.13\% in classifying 15 various controllers based on only three features. The experiment demonstrated that the suggested method can categorize the same make and model of UAV controllers without reducing overall accuracy.

In [86], the authors developed a mechanism for detecting UAVs with both an RC controller and Wi-Fi or Bluetooth-linked mobile applications based on radio communication and an internet protocol (IP) address. The detection system was divided into two stages. The first stage involves searching for the presence of any unfamiliar IP address in the range of interest. As the UAV has a Wi-Fi connection, it has its own IP address. The drone detection system employs an algorithm that stores certain IP addresses of known devices in its database. Then, utilizing a Wi-Fi adaptor, the algorithm will continue to verify whether any unfamiliar device is connected to the router by matching its IP address with those stored in the database. In the event that any unfamiliar device is connected to the router, the system outputs its IP address. The second stage entails determining whether the detected unfamiliar device is an UAV or not by using an RF receiver. Overall, the detection system prototype could be able to detect drone presence in the range of 100 meters, even when birds appear in the region of interest.

Comment 2: [The performance discussion on the battery life in this work can be added.]

Response: Thank you! We found your comments extremely helpful and have revised 1.2 section “Key challenges in UAV detection” accordingly by discussing about UAV performance based on battery lifetime .

5) Battery life. One of the biggest issues that UAVs encounter is their limited battery life. Increasing the battery size of UAVs is not possible since it would increase weight, which is another key consideration. Most research concluded that lithium polymer (LiPo) batteries are the most commonly utilized battery type in drones; nevertheless, lithium iron phosphate (LiFePO4) batteries are believed to be safer and have a longer life cycle [90, 91]. Drones can fly in the air for 30–40 minutes before landing, and swapping batteries might be troublesome. Temperature, humidity, altitude, and flight speed all have a major influence on drone batteries. However, the charging strategies for these batteries are equally critical. The right charging protocol is crucial for extending battery life and ensuring consistent performance. To mitigate the impact of UAV flight time limitations imposed by limited on-board battery life, it is critical to reduce energy consumption in UAV operations [90].

Comment 3: [The acronyms (eg., RF) should be explained when they first appear, even in Abstract.]

Response: Thank you very much for the reminder. We have made revisions accordingly:

radio frequency (RF)—line 24.

Internet of Things (IoT)—line 27

Wireless fidelity (Wi-Fi): line 27

Artificial Intelligence (AI) (line 181)

frequency modulation (FM) radio, line 210.

Institute of Electrical and Electronics Engineers (IEEE) (line 219)

Convolutional Neural Network (CNN) (line 238)

Long-short-term memory-adaptive learning rate optimizing (LSTM-ALRO) (line 245)

Helicopter Rotor Modulation (HERM) (line 293)

Global Positioning System (GPS): line 323

eXtreme Gradient Boosting (XGBoost) (line 357)

k-nearest neighbors (KNN): line 357

Comma-Separated Values (CSV): line 360

false negative rate (FNR) (line 369)

false discovery rate (FDR) (line 369)

adaptive boosting (AdaBoost) (line 374)

multilayer perceptron (MLP) (line 375)

deep neural network (DNN)—line 397

Visual Geometry Group (VGG): lines 404–405

1-dimensional convolutional neural network (1D-CNN) (lines 426-427)

Support vector machine (SVM) (lines 543-544)

receiver operating characteristic (ROC)—line 522

Zero-Crossing Rate (ZCR) (line 530)

Convolutional recurrent neural network (CRNN) (lines 580–581)

Convolutional Long Short-Term Memory (CLSTM) (lines 621-622)

Gated Recurrent Unit (GRU): line 658

Bidirectional Long short-term memory (BiLSTM) (line 658)

Region-Based Convolutional Neural Network (R-CNN) (line 682)

Detection Transformer (DETR): line 693

Common Objects in Context (COCO): line 695

average precision (AP)—line 698

Intersection over Union (IoU): lines 698–699.

pan, tilt, zoom (PTZ)—line 703

residual network (ResNet)—line 761

mean average precision (mAP)—line 763

medium access control (MAC)—line 808.

Comment 4: [It is better to resize the oversized figures.]

Response: Agree. We have resized all the figures accordingly.

Comment 5: [The interference factors in real-world scenarios can be discussed.]

Response: We think this is an excellent suggestion. We have incorporated your comment and tried to add the suggested content to Table 7 with the name "Real-world interference factors in drone detection technologies” in the revised manuscript.

Lines:

The accuracy of any UAV detection system might be hindered by different interference factors in a real-world scenario. Table 7 lists the main examples of real-life interferences for each detection technology, respectively.

Comment 6: [Go through the paper carefully, and remove the typos and formatting issues (e.g., “3) Pulse Doppler Radar:” -> “3) Pulse doppler radar:”?).]

Response: Thank you for your nice reminder. We have gone through the entire manuscript carefully to eliminate typos and other mistakes.

Comment 7: [Make the References more comprehensive, besides this work, some other promising scenarios (e.g., Big data, other IoT systems) can be covered in this work. If the above related work can be discussed, it can strongly improve the research significance. For the improvement, the following papers can be considered to make the references more comprehensive.

  1. Wang, C. Jin, Q. Tang, N. N. Xiong and G. Srivastava, "Intelligent Ubiquitous Network Accessibility for Wireless-Powered MEC in UAV-Assisted B5G," in IEEE Transactions on Network Science and Engineering, vol. 8, no. 4, pp. 2801-2813, 1 Oct.-Dec. 2021, doi: 10.1109/TNSE.2020.3029048.]

Response: Response: Thank you very much for your valuable feedback! Accordingly, throughout the manuscript, we have revised Section 2 by adding the extra subsection 2.5 containing suggested content about sensor fusion technologies, drone detection and identification based on Wi-Fi wireless communication, as well as cellular and IoT networks.

P/S: Dear Reviewer, first of all, we would like to thank you for your extra valuable time, attention, and kindly response to our research and the corresponding manuscript. We do thank you very much for your detailed feedback, suggestions, and comments on our manuscript.  Indeed, your comments were immensely valuable and helpful.  We hope the manuscript, after careful revisions, meets your high standards. If not, we apologize in advance. The authors welcome further constructive comments, if any. Thank you very much.

Sincerely, authors

Round 2

Reviewer 1 Report

Comments and Suggestions for Authors

1) statistical support on threats and incidents:

Response: We think this is an excellent suggestion. We have incorporated your comment and condensed the text content about drone threats and incidents by indicating the statistical support for recent drone incidents in Table 1.

2nd review commentTable 1 presents a compilation of individual incidents, amounting to a total of 10 incidents, in a tabular format. However, it should be noted that this table does not constitute statistical support. A comprehensive statistical analysis requires inclusion of elements such as the time period, total number of incidents, categories, and the percentage for each category. Additionally, it is crucial to incorporate a reference to the source of the data for transparency and credibility.

Comments on the Quality of English Language

Moderate editing of English language required

Author Response

Dear Dr. Reviewer,

Thank you for giving us the opportunity to submit a revised draft of the manuscript titled “Advances and Challenges in Drone Detection and Classification Techniques: A State-of-the-Art Review” to “MDPI Sensors." We appreciate your precious time and effort in reviewing our manuscript and providing valuable feedback. We are also grateful for your immensely helpful suggestions and insightful comments that led to possible improvements in the current version. The comments are reviewed carefully, and the point-by-point response to each individual comment is provided in the following:

All modifications in the manuscript have been highlighted in blue.

1) statistical support on threats and incidents:

2nd review comment: Table 1 presents a compilation of individual incidents, amounting to a total of 10 incidents, in a tabular format. However, it should be noted that this table does not constitute statistical support. A comprehensive statistical analysis requires inclusion of elements such as the time period, total number of incidents, categories, and the percentage for each category. Additionally, it is crucial to incorporate a reference to the source of the data for transparency and credibility.

Response: Agreed. Thank you for pointing this suggestion out. Accordingly, throughout the manuscript, we have revised Section 1.1, “Significance of UAV Detection: Drone Threat Categories and Incidents,” by adding the suggested content:

  1. The number of total drone incidents (lines 117–118), as well as the number and percentage of the four main drone threat categories (Figure 2)
  2. The number of four main drone threat categories by month, starting from January 2023 until the end of November 2023 (Figure 3),
  3. The percentage of individual threat categories by country: lines 119–137

The article has been thoroughly checked for grammatical and punctuation mistakes.

P/S: Dear Reviewer, first of all, we would like to thank you for your extra valuable time, attention and kindly response to our research and the corresponding manuscript. We do thank you very much for your detailed feedback, suggestions and comments on our manuscript.  Indeed, your comments were immensely valuable and helpful.  We hope the manuscript after careful revisions, meets your high standards. If not, we apologize in advance. In this regard, we would also like to mention the problem of lack of time, and we intend to use some of your recommendations in future works. Thank you very much.

Sincerely, authors
